# CORRECT: Condensed Error Recognition via Knowledge Transfer in Multi-agent Systems

## Abstract

Multi-agent systems (MAS) are increasingly capable of tackling complex real-world tasks, yet their reliance on inter-agent coordination, tool use, and long-horizon reasoning makes error recognition particularly challenging. Minor errors can propagate across agents, escalating into task failures while producing long, intertwined execution trajectories that impose significant costs for both human developers and automated systems to debug and analyze. Our key insight is that, despite surface differences in failure trajectories (e.g., logs), MAS errors often recur with similar structural patterns. This paper presents *CORRECT*, the first lightweight, training-free framework that leverages an online cache of distilled error schemata to recognize and transfer knowledge of failure structures across new requests. This cache-based reuse allows LLMs to perform targeted error localization at inference time, avoiding the need for expensive retraining while adapting to dynamic MAS deployments in subseconds. To support rigorous study in this domain, we also introduce *CORRECT*-Error, a large-scale dataset of over 2,000 annotated trajectories collected through a novel error-injection pipeline guided by real-world distributions, and further validated through human evaluation to ensure alignment with natural failure patterns. Experiments across seven diverse MAS applications show that CORRECT improves step-level error localization up to 19.8% over existing advances while at near-zero overhead, substantially narrowing the gap between automated and human-level error recognition.

## 1 Introduction

Multi-agent systems (MAS) have emerged as a powerful paradigm for solving complex tasks that require diverse capabilities and collaborative problem-solving, with demonstrated success in various domains, including software development (Qian et al., 2023; Hong et al., 2023; Zhang et al., 2024), scientific research (Lu et al., 2024), web navigation (Zhou et al., 2023), and general-purpose task automation (Wu et al., 2024; Wang et al., 2025). By orchestrating multiple specialized agents, MAS can tackle challenges beyond the reach of single-agent systems (SAS) that rely on single LLMs for task solving, achieving human-level performance on benchmarks like GAIA (Mialon et al., 2023) and SWE-bench (Jimenez et al., 2023).

However, as MAS increasingly scale in both complexity (e.g., sophisticated interactions with other agents and tools) and deployments, decisive error recognition in MAS deployments becomes exceptionally challenging. Unlike SAS where failures can be traced to a single faulty output, MAS failures often emerge from cascading effects across agents: a single error-prone step by one agent can propagate through downstream interactions, ultimately causing task failure (Gao et al., 2025). We refer to this fundamental challenge as *decisive error recognition*: pinpointing the precise agent and step that first triggered the failure (Zhang et al., 2025). Efficient decisive error recognition is essential for reliable MAS deployments, sustaining service quality, and guiding operational management such as safe agent upgrades, targeted restarts, and service monitoring (Epperson et al., 2025).

Unfortunately, decisive error recognition in MAS remains open-ended due to three fundamental obstacles: (1) *Generality*: MAS span a wide spectrum of applications, and even within an application, requests can exhibit drastically different error patterns, making it difficult to design methods that generalize. Existing advances (Zhang et al., 2025) often resort to LLM-as-a-judge methods for generality, yet achieve $\leq 10\%$ accuracy in pinpointing the exact error step, barely above the accuracy

of random guessing (3%). Recent efforts (Ge et al., 2025) to improve accuracy through fine-tuning not only hurt generalization, but fall short in (2) *Data Efficiency*: obtaining labeled data in error recognition is notoriously expensive and inherently ambiguous. For example, in the WHO&WHEN dataset (Zhang et al., 2025), annotators spent over 30 expert hours labeling fewer than 200 trajectories, yet disagreement rates exceeded 50%. This makes any training-based approaches ineffective without voluminous data; and (3) *Computation Efficiency*: even with sufficient data, tuning LLMs for error recognition may be impractical to catch up with deployments where new error types arise continuously and sporadically, e.g., new attacks in cloud AIOps (Wang et al., 2025).

In this paper, we first notice that failures in MAS tend to recur with similar structures across requests (§2.2). Because a MAS application often relies on the same role specifications, orchestration rules, tool APIs, and verification policies, diverse requests often funnel into common decision skeletons. Our real-world analysis of WHO&WHEN dataset proposed in Zhang et al. (2025) shows that over 80% of failure trajectories have at least one counterpart with $\geq 0.8$ semantic similarity in error logs. This suggests that error knowledge can be systematically *distilled, cached, and reused*. However, naive approaches, such as in-context learning (ICL) that insert prior trajectories as in-context exemplars, quickly break down: logs can exceed 32,000 tokens (Yang et al., 2025; Dubey et al., 2024), contain low-entropy noise, and even underperform zero-shot baselines (§2.2). These challenges demand a new approach: one that is general, data-efficient, and lightweight for practical deployments.

In this paper, we propose *CORRECT* (COndensed eRror RECognition via knowledge Transfer), a novel framework that formalizes decisive error recognition as a first-class problem for reliable MAS, and automatically distills prior error patterns into compact, reusable schemas that capture their essential signatures, triggering contexts, and propagation patterns. During inference, *CORRECT* identifies relevant schemas and applies them to the new request, enabling accurate, lightweight, and training-free recognition in dynamic environments. Our contributions are summarized as follows:

- **CORRECT: the first schema-guided detector with broad generality.** We introduce *CORRECT*, the first framework that distills recurrent MAS failures into compact error schemata and reuses them for decisive error recognition. Unlike LLM-as-a-judge approaches that trade accuracy for generality, or costly fine-tuning approaches, *CORRECT* leverages schemata to transfer knowledge across requests in real time. This design improves step-level localization accuracy up to 20 points over state-of-the-art designs (Zhang et al., 2025), while remaining training-free and computationally efficient, making it practical for deployment in dynamic MAS environments.

- *CORRECT*-**Error: a versatile, large-scale, and high-fidelity dataset for MAS error recognition.** To close the benchmarking gap, we construct *CORRECT*-Error, a large-scale collection of over 2,000 multi-agent trajectories with fine-grained, step-level error annotations. Unlike prior datasets that are small and noisy, *CORRECT*-Error is built using a novel error-injection pipeline guided by real-world failure distributions, yielding trajectories that balance controlled coverage with the realism of natural MAS failures. We conduct extensive human validation, confirming strong alignment between synthetic annotations and expert judgment. *CORRECT*-Error not only enables rigorous evaluation of *CORRECT*, but also establishes a reusable, extensible benchmark for the community to advance the development of MAS error recognition methods.

Together, these contributions advance the state of the art in MAS error recognition and establish a new foundation and datasets for making MAS more reliable, interpretable, and deployable at scale.

## 2 BACKGROUND AND MOTIVATION

### 2.1 DECISIVE ERROR IN MULTI-AGENT SYSTEMS

Task failures in MAS often arise from specific *decisive errors* that, once committed, render successful task completion impossible. We formalize this notion following prior advances (Zhang et al., 2025). Consider a MAS executing a trajectory $\tau = \{(a_1, s_1), (a_2, s_2), \ldots, (a_T, s_T)\}$, where agent $a_i$ performs step $s_i$. The outcome of the trajectory is denoted by $\mathcal{R}(\tau) \in \{0, 1\}$, with 1 for success and 0 for failure. A step $(a_k, s_k)$ in a failed trajectory $\tau$ is a *decisive error* if replacing it with a correct alternative $\tilde{s}_k$ would change the outcome to success. Formally, the earliest decisive error is: $(a^*, s^*) = \min_{k \in \mathcal{D}(\tau)} k$, where $\mathcal{D}(\tau) = \{ k : \mathcal{R}(\tau) = 0 \wedge \mathcal{R}(\tau_{[s_k \to \tilde{s}_k]}) = 1 \}$, and $\tau_{[s_k \to \tilde{s}_k]}$ denotes the modified trajectory in which step $s_k$ is replaced by $\tilde{s}_k$.

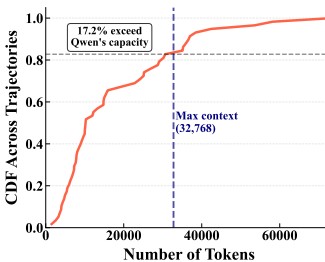 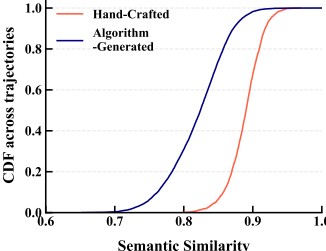 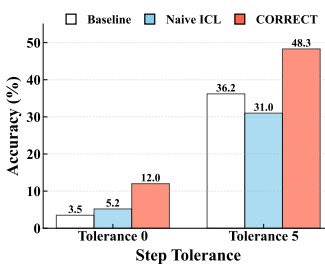

Figure 1: MAS failure traces are complex and long, often exceeding the model capacity.

Figure 2: Failure trajectories exhibit high semantic similarities (Who and When dataset).

Figure 3: Performance of naive ICL and our method (Who and When dataset).

Intuitively, a decisive error is **the earliest step** whose correction flips the trajectory outcome from failure to success. Identifying decisive errors is fundamental to reliable MAS. Unlike coarse-grained error recognition, which only flags failed trajectories, decisive error recognition pinpoints the exact *agent* and *step* responsible for initiating failure. This precision enables targeted interventions such as tuning role specifications, refining orchestration logic, or upgrading individual agents, without costly overhauls of the entire system.

## 2.2    MOTIVATIONS OF *CORRECT*

Automated decisive error recognition in MAS has primarily followed two directions: LLM-as-a-judge approaches (Zheng et al., 2023; Zhang et al., 2025) and fine-tuning specialized LLMs (Cemri et al., 2025). Both exhibit fundamental limitations in accuracy, generality, and efficiency.

**Limitations of Existing Advances.**    LLM-as-a-judge methods were initially designed to rate the quality of LLM outputs, achieving up to 80% agreement with human preferences (Zheng et al., 2023). Zhang et al. (2025) extended this paradigm to MAS error attribution, introducing three variants: (i) *all-at-once*, which provides the full error log to the LLM and asks it to identify the responsible agent and error step; (ii) *step-by-step*, which incrementally reveals the trajectory and checks errors at each step; and (iii) *binary search*, which recursively partitions the log to localize the error. However, as MAS becomes more complex, which elongates the failure trajectory, these methods lose diagnostic precision (Figure 1): on the WHO&WHEN dataset (Zhang et al., 2025), Qwen-2.5-7B achieves only 3.5% step-level accuracy, far below practical deployment requirements.

Fine-tuning (FT)-based approaches, whether supervised or reinforcement learning–based, face substantial efficiency and expense challenges. Their success hinges on large-scale, high-quality labeled datasets. Yet annotating MAS failures is prohibitively expensive: annotators must disentangle long, interdependent interactions across agents and tools, taking 30 expert hours for annotating fewer than 200 trajectories. Worse, error trajectories vary across applications and requests, making FT-trained detectors brittle and undermining generalization for diverse deployments at scale.

**Pervasive Error Similarity Yet Hard to Reuse.**    Despite these challenges, our analysis of the WHO&WHEN dataset (Figure 2) reveals that more than 80% of failed requests share a semantic (cosine) similarity above 0.8, measured via BERT-based embeddings. This reveals an underexploited opportunity: historical failures could be reused as exemplars for new requests. A natural attempt is to adopt in-context learning (ICL), retrieving and appending similar trajectories to guide error recognition. However, our experiments (Figure 3) show that such a strawman approach even degrades recognition accuracy, due to two primary limitations: (i) *extreme trajectory length*: 17% of trajectories exceed 32K tokens, surpassing the context length of most LLMs (e.g., Qwen3 (Yang et al., 2025)); and (ii) *low signal-to-noise ratio*: execution trajectories interleave request-specific details and tool outputs with the true error-inducing steps, diluting critical information.

## 3    CORRECT: CONDENSED ERROR RECOGNITION VIA KNOWLEDGE TRANSFER

Our observations call for a novel approach that can *systematically reuse structural error knowledge* without overwhelming context or succumbing to noise, while remaining general, data-efficient, and

lightweight for real-time MAS deployment. To these ends, we introduce *CORRECT* (**CO**ndensed e**R**ror **REC**ognition via knowledge **T**ransfer), the first framework that distills past errors into compact error schemas and adaptively applies them for accurate decisive error recognition without any training, enabling efficient adaptation to diverse tasks and errors. As summarized in Algorithm 1, CORRECT combines three interconnected phases: (1) *Offline schema extraction*, (2) *Online schema-guided error recognition*, and (3) *Dynamic schema management*.

## 3.1 ERROR SCHEMA EXTRACTION

Given an annotated error trajectory $\mathcal{T} = \{(a_i, s_i, r_i)\}_{i=1}^n$, where $a_i$ denotes the agent at step $i$, $s_i$ represents the step content, and $r_i$ is the corresponding result, along with the identified error at step $s_e$ and error reason $r_e$, *CORRECT* generates an error schema $\mathcal{S}$ capturing (Figure 4): (1) *Error Signatures* $\Sigma$: Characteristic patterns such as agent actions, interaction sequences, and key behavioral markers; (2) *Error Context Analysis* $\mathcal{C}$: Detailed analysis of the conditions that led to the error, including agent states, task progress, and environmental factors; and (3) *Detection Heuristics* $\mathcal{H}$: Actionable rules and guidelines for identifying similar errors in new contexts.

To minimize human efforts, *CORRECT* leverages capable LLMs (e.g., GPT-5) to generate error schemas. We discuss how to ensure the quality of the schema in Section 3.2. A detailed comparison between *CORRECT* 's automatically distilled schemata and human-curated taxonomies such as MAST are provided in Appendix A.8.

**Clustered Schema Extraction.** Even with LLMs, generating a schema for every trajectory is still costly due to voluminous requests in practical MAS deployments. In fact, doing so is unnecessary because schema reuse often follows a long-tailed distribution: a small number of schemas are frequently reused, while most are rarely applied (§5). To exploit this, *CORRECT* performs

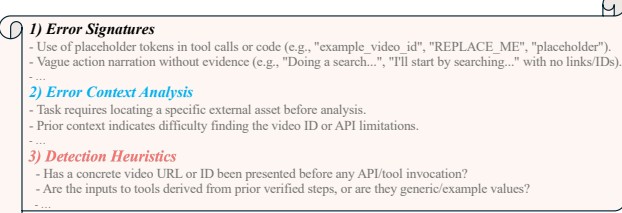

Figure 4: Example of an error schema generated on the WHO&WHEN dataset.

the following offline procedure: (1) *Trajectory Clustering:* Failure trajectories are embedded semantically and clustered to group similar error patterns, and then (2) *Cluster-level Schema Generation:* One representative schema is generated per cluster, capturing the common error structure without redundant costs.

## 3.2 SCHEMA-GUIDED ERROR RECOGNITION ONLINE

Once a failure request requires diagnosis, we start with its trajectory $\mathcal{T}_{\text{target}}$ and retrieve the top-k relevant schemas from the cache via semantic similarity search (e.g., cosine similarity of embeddings):

$$\text{sim}(\mathcal{T}_{\text{target}}, \mathcal{S}_i) = \cos(\text{embed}(\mathcal{T}_{\text{target}}), \text{embed}(\mathcal{S}_i))$$

With retrieved schemas, we prompt the LLM to perform error recognition by instantiating the schemas in the context of the target trajectory. These schemas, together with the trajectory and lightweight adaptation instructions, are passed to an LLM for diagnosis. Formally, the LLM receives $(\mathcal{T}_{\text{target}}, \{\mathcal{S}_j\}_{j=1}^k, \text{prompt}_{\text{detect}})$ as input prompt and produces the error recognition result:

$$\text{result} = \text{LLM}_{\text{detect}}\left(\mathcal{T}_{\text{target}}, \{\mathcal{S}_j\}_{j=1}^k, \text{prompt}_{\text{detect}}\right).$$

By leveraging schemas as condensed expert knowledge, this process guides the LLM to focus on salient failure patterns without the overhead of processing entire historical trajectories.

**Adaptation with Schema Expansion and Distillation.** *CORRECT* maintains an effective schema cache through two complementary mechanisms. First, *schema expansion*: when user feedback confirms successful recognition, *CORRECT* leverages the ground truth label from the user to generate and cache a new error schema following 3.1. Priority is given to trajectories with low similarity to existing schemas (i.e., $\text{sim}(\mathcal{T}_{\text{new}}, \mathcal{S}_i) < \delta$ for all cached schemas), ensuring the cache captures diverse

---

**Algorithm 1:** CORRECT Framework

---

**Input:** Annotated trajectories $\{(\mathcal{T}, s_e, r_e)\}$ for schema extraction; target trajectory $\mathcal{T}_{\text{target}}$ for diagnosis
**Output:** Error recognition result $(a^*, s^*, c)$

**Offline Schema Extraction** (Sec 3.1)
Cluster trajectories by semantic similarity and generate one representative schema per cluster;
**foreach** *annotated trajectory* $(\mathcal{T}, s_e, r_e)$ **do**
    Generate schema: $\mathcal{S} \leftarrow \text{LLM}_{\text{extract}}(\mathcal{T}, s_e, r_e)$ ;
    Apply filtering/distillation to enforce compactness, token filtering, and consistency;
    Insert into cache: $\mathcal{C}.\text{put}(\mathcal{S})$ ;

**Online Schema-Guided Error Recognition** (Sec 3.2)
Embed target trajectory: $\mathbf{e} \leftarrow \text{embed}(\mathcal{T}_{\text{target}})$ ;
Retrieve top-$k$ schemas: $\{\mathcal{S}_j\}_{j=1}^{k} \leftarrow \mathcal{C}.\text{search\_top\_k}(\mathbf{e})$ ;
Update access statistics: $\forall j : \mathcal{C}.\text{update\_access}(\mathcal{S}_j)$ ;
Diagnose decisive error: $(a^*, s^*, c) \leftarrow \text{LLM}_{\text{detect}}(\mathcal{T}_{\text{target}}, \{\mathcal{S}_j\}, \text{prompt}_{\text{detect}})$ ;

**Dynamic Schema Management** (Sec 3.2)
**if** *user feedback confirms successful recognition* **then**
    **if** $sim(\mathcal{T}_{target}, \mathcal{S}_i) < \delta$ *for all* $\mathcal{S}_i \in \mathcal{C}$ **then**
        Extract ground truth label from data;
        Distill new schema $\mathcal{S}_{\text{new}}$ from $\mathcal{T}_{\text{target}}$ with ground truth;
        $\mathcal{C}.\text{put}(\mathcal{S}_{\text{new}})$;

**if** $\mathcal{C}.access\_count(\mathcal{S}_j) > \theta_{hot}$ **then**
    Generate candidates: $\{\mathcal{S}_i'\}_{i=1}^{m} \leftarrow \text{LLM}_{\text{extract}}^{m}(\mathcal{T}_j, s_{e,j}, r_{e,j})$ ;
    Evaluate by replaying on prior trajectories;
    Select best schema: $\mathcal{S}^* \leftarrow \arg\max_{\mathcal{S}_i'} \text{accuracy}(\mathcal{S}_i')$ ;
    Replace old schema: $\mathcal{C}.\text{replace}(\mathcal{S}_j, \mathcal{S}^*)$ ;
**return** $(a^*, s^*, c)$

---

error patterns rather than redundant ones. We note that human supervision is naturally abundant in practical deployments, and we include an extended analysis of validation strategies and related cache-based approaches in Appendix A.9.

Second, *schema distillation*: expansion alone may yield suboptimal quality, and frequently accessed error schemas (cache hits $> \theta_{\text{hot}}$) may benefit from further refinement. In such cases, *CORRECT* generates multiple candidate schemas and replays them against prior trajectories to select the discriminative one with the highest accuracy.

Together, expansion ensures coverage for novel errors online, while distillation preserves cache efficiency by retaining only high-quality, discriminative schemas. As shown in algorithm 1, the offline stage (Sec 3.1) builds an initial cache by distilling clean, representative schemas from annotated trajectories. At test time (Sec 3.2), the system retrieves the top-$k$ relevant schemas to guide the LLM detector toward the most likely error. The cache tracks how often each schema is used so it can update itself: new schemas are added when no good match exists, and frequently accessed schemas are refined when needed. This keeps the cache compact, accurate, and aligned with live error patterns.

We further evaluate CORRECT's behavior under evolving request distributions, showing that the schema cache adapts robustly over time (Section 5.2).

# 4 *CORRECT*-ERROR: A LARGE-SCALE ERROR DETECTION BENCHMARK

Existing efforts for trajectory-level error analysis are limited in both scale and diversity, and human annotation is costly and difficult to scale (§2.2). To bridge this gap and evaluate the effectiveness of *CORRECT* (§3), we introduce *CORRECT*-Error, a large-scale benchmark that faithfully reflects the distribution of natural errors encountered in real-world MAS.

### 4.1 BOOTSTRAP ERROR SYNTHESIS PIPELINE

In building *CORRECT*-Error, we develop a bootstrap methodology that uses a small set of human-annotated error trajectories as seeds for scalable error generation, blending realism with controllability. It follows a three-stage pipeline that distills human expertise into scalable synthetic data while preserving the structural and semantic integrity of real-world error patterns.

**Stage 1: Diverse Trajectory Collection.** We begin by generating a large pool of successful multi-agent trajectories spanning heterogeneous tasks and domains. Alongside, we curate a smaller but high-quality set of human-annotated error trajectories. These serve as reference exemplars of realistic failure dynamics, capturing both localized mistakes and their downstream propagation.

**Stage 2: Semantic Error Schema Matching.** Each successful trajectory is paired with its closest human-labeled error trajectory using semantic similarity measures that account for both high-level task goals and fine-grained agent interactions. This alignment ensures that the selected error schema is contextually aligned with the target trajectory, avoiding unrealistic mismatches. Then we use GPT-5 to devise an error injection strategy that specifies (i) where in the target trajectory to introduce the error, and (ii) how to adapt the error pattern while preserving its core semantics.

**Stage 3: Contextual Error Injection.** Following the injection strategy, we prompt GPT-5 that generated the original successful trajectory to introduce an erroneous action at the designated point. This guarantees consistency in linguistic and behavioral style while embedding a realistic failure. We provide the prompt used to generate natural, schema-consistent error injections in Appendix A.11.

### 4.2 HUMAN-ALIGNMENT ANALYSIS

Following our bootstrap pipeline (§4.1), we synthesized over 2,000 trajectories across seven datasets (Figure 5), yielding **12.3×** more data than WHO&WHEN with cost over 3 billion tokens using GPT-5 series models and GPT-4o series models based on Magnetic-One (Fourney et al., 2024) and AutoGen (Wu et al., 2024). The resulting benchmark spans diverse tasks, including multi-hop QA, common planning, mathematical reasoning, and scientific problem-solving. By leveraging limited human annotations as seeds, our novel pipeline generates diverse error scenarios at scale.

To rigorously assess this authenticity, we conducted a human evaluation study involving three independent expert labelers over 120 hours. Each labeler was shown an equal mix of synthetic and human-labeled error trajectories, without disclosure of their origin. As shown in Figure 5, labelers struggled to distinguish between the two: 47.1% of synthetic trajectories were misclassified as human-labeled, while 42.3% of genuine trajectories were correctly

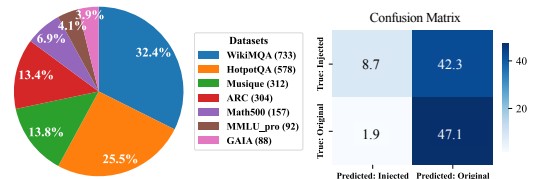

Figure 5: *CORRECT*-Error includes diverse tasks. The synthesized data preserves high realism, where human labelers frequently misclassified synthetic errors as genuine ones.

identified. This near-random classification accuracy (close to 50% in both cases) indicates that our synthetic errors are effectively indistinguishable from real ones. Moreover, we notice that a high inter-annotator agreement on authenticity: 94.4% of synthetic trajectories were judged as genuine by at least two out of three labelers, with 52.9% receiving unanimous consensus. We add detailed human-alignment analysis in Appendix A.4.

Together, these results validate that our error injection pipeline faithfully captures the nuanced characteristics of real-world MAS failures. We emphasize that this methodology enables data generation that is *cheap* (automated and low-cost), *abundant* (scalable to millions of examples), *precise* (with unambiguous ground-truth labels), and *realistic* (closely matching natural error distributions derived from human annotations), building the foundation for training effective recognition models.

## 5 EXPERIMENTS

We demonstrate that *CORRECT* significantly enhances error recognition, achieving improvements of up to 20% on WHO&WHEN and an average gain of 28.7% across MAS tasks (§5.1) on CORRECT-ERROR, all at zero training costs. *CORRECT* remains robust under distribution shifts arising from model updates, dataset variations, schema cache size, and the number of stored schemata (§5.3). We provide case-studies with illustrative example and detailed analysis in Appendix A.13

**Models and Tasks.** We evaluate *CORRECT* on both the human-annotated WHO&WHEN benchmark and our high-quality benchmark, CORRECT-ERROR, spans diverse tasks including multi-hop QA (HotpotQA (Yang et al., 2018), Musique (Trivedi et al., 2022), WikiMQA (Ho et al., 2020)), scientific reasoning (ARC (Clark et al., 2018), MMLU-Pro (Wang et al., 2024)), mathematical reasoning (Math500 (Lightman et al., 2023)), and general agentic tasks including planning (GAIA (Mialon et al., 2023)). WHO&WHEN consists of two subsets: a Human-Crafted subset and an Algorithm-Generated subset. Experiments are conducted on both open- and closed-source models, including the Qwen (Yang et al., 2024; 2025), Llama (Dubey et al., 2024), GPT (Hurst et al., 2024; OpenAI, 2025), DeepSeek-R1 (Guo et al., 2025), and Gemini series (Comanici et al., 2025). We mask each trajectory itself and avoid receiving its own error schema for preventing data leakage. Additional experimental details are provided in Appendix A.3.We include the SFT details and hyperparameters in Appendix A.10, and experiments on Error-Category Detection Cemri et al. (2025) in Appendix A.14.

**Baselines.** We compare *CORRECT* against three state-of-the-art approaches:

- *LLM-as-a-Judge*, a zero-shot prompting strategy where an LLM directly inspects trajectories without auxiliary guidance (Zhang et al., 2025; Peng et al., 2023a);

- *Fine-tuning*, where an LLM is trained on the full trajectory dataset with cross-entropy loss to encode domain-specific failure patterns (Chen et al., 2025; Fu et al., 2025); and

- *Naive In-Context Learning*, which inserts complete error trajectories as few-shot exemplars, which may be constrained by limited context windows (Dong et al., 2022; Yu et al., 2025).

- *MIPRO*, which optimizes prompts and few-shot demonstrations via Bayesian search over instruction–example combinations using success signals from prior runs. (Opsahl-Ong et al., 2024)

**Metrics.** Following existing advances (Zhang et al., 2025), we report *step-level accuracy*, which provides actionable debugging signals. To account for the ambiguity of error attribution, we additionally report accuracy@k, where predictions within $k$ steps of the ground truth are treated as correct, e.g., Acc@0 requires identifying the exact erroneous step, while Acc@1 tolerates an offset of one step. This better reflects practical debugging scenarios, where approximate localization is often sufficient. We report the average performance over five independent runs.

### 5.1 END-TO-END PERFORMANCE

**CORRECT achieves significant gains in error recognition accuracy (WHO&WHEN dataset).** Table 1 shows that *CORRECT* consistently surpasses existing advances across both human-crafted and algorithm-generated subsets. On human-crafted data, *CORRECT* raises Qwen-2.5-7B's exact-step accuracy from 3.5% to 12.1% (a $3.5\times$ improvement), and improves GPT-5 from 8.6% to 17.2%. These gains extend to tolerant metrics as well, with GPT-5 + *CORRECT* achieving 37.8% at Acc@1 versus 24.1% for the baseline. By contrast, fine-tuning (3.5%) and naive ICL (5.2%) offer only marginal improvements, suggesting that standard supervised learning and raw in-context trajectories fail to capture complex error patterns. On algorithm-generated data, *CORRECT* maintains clear advantages, with Gemini-2.5-Flash improving from 31.8% to 38.9% (+7.1 points) and GPT-5 exhibiting the largest gain (+19.8 points). These consistent improvements across model families (Qwen, Gemini, GPT) and scales (7B to GPT-5) highlight the effectiveness of condensed error schemas.

**CORRECT delivers 17–28% average improvements (CORRECT-ERROR benchmark).** Table 2 highlights *CORRECT*'s strong generalization across seven datasets. For GPT-4o-mini subset, *CORRECT* improves average accuracy by +20.1%, +27.6%, and +28.7% at Acc@1, Acc@3, and Acc@5, respectively using Qwen-2.5-7b. Gains are especially pronounced on knowledge-intensive tasks: HotpotQA (+26.1 points), WikiMQA (+29.4 points), and Math500 (+46.9 points). At higher toler-

| Method | Model | Human-Crafted | | Algorithm-Generated | |
|---|---|---|---|---|---|
| | | Acc@0 | Acc@1 | Acc@0 | Acc@1 |
| LLM-as-a-Judge | Qwen-2.5-7b | 3.5 | 8.6 | 19.1 | 42.9 |
| | Qwen-3-30b | 1.7 | 5.2 | 15.1 | 42.9 |
| | Qwen-3-80b | 6.9 | 8.6 | 21.4 | 47.6 |
| | Llama-8b | 1.7 | 3.5 | 3.2 | 15.9 |
| | DeepSeek-R1 | 3.5 | 17.2 | 23.8 | 54.0 |
| | Gemini-2.5-flash | 5.2 | 13.8 | 31.8 | 56.0 |
| | Gemini-2.5-pro | 3.5 | 8.6 | 25.4 | 50.0 |
| | GPT-4o-mini | 3.5 | 12.5 | 12.7 | 39.7 |
| | GPT-4o | 3.5 | 10.3 | 18.3 | 50.0 |
| | GPT-5-nano | 1.7 | 5.2 | 19.1 | 41.3 |
| | GPT-5 | 8.6 | 24.1 | 18.3 | 56.4 |
| Fine-tuned LLM | Qwen-2.5-7b | 3.5 | 11.9 | 18.9 | 42.9 |
| Naive ICL | Qwen-2.5-7b | 5.2 | 10.3 | 15.9 | 40.5 |
| MIPRO | Qwen-2.5-7b | 8.6 | 12.1 | - | - |
| *CORRECT* | Qwen-2.5-7b | 12.1 (+8.6) | 15.5 (+6.9) | 19.8 (+0.7) | 46.8 (+3.9) |
| | Gemini-2.5-flash | 10.3 (+5.1) | 20.7 (+6.9) | 38.9 (+7.1) | 55.2 (-0.8) |
| | Gemini-2.5-pro | 5.2 (+1.7) | 15.5 (+6.9) | 24.6 (-0.8) | 52.4 (+2.4) |
| | GPT-5-nano | 6.9 (+5.2) | 17.2 (+12.0) | 24.6 (+5.5) | 44.4 (+3.1) |
| | GPT-5 | **17.2 (+8.6)** | **37.8 (+13.7)** | 38.1 (+19.8) | **58.8 (+2.4)** |

Table 1: *CORRECT* achieves higher error recognition accuracy over existing advances on WHO&WHEN dataset. Acc@0 denotes exact-step accuracy (the model must pinpoint the precise error step). Acc@k denotes tolerant accuracy (a prediction is correct if it falls within $\pm k$ steps of the ground truth). For rows corresponding to *CORRECT*, numbers in parentheses indicate absolute improvements over the LLM-as-a-Judge baseline.

| Method | Tolerance | Dataset | | | | | | | Avg. |
|---|---|---|---|---|---|---|---|---|---|
| | | Gaia | HotpotQA | Musique | WikiMQA | Arc | Math500 | MMLU-Pro | Improv. |
| Synthesized by **GPT-4o-mini** | | | | | | | | | |
| LLM-as-a-Judge | Acc@1 | 28.6 | 34.8 | 27.8 | 14.7 | 64.0 | 10.2 | 58.3 | - |
| | Acc@3 | 42.9 | 59.4 | 77.8 | 55.9 | 75.0 | 23.4 | 62.5 | - |
| | Acc@5 | 50.0 | 63.8 | 77.8 | 64.7 | 78.0 | 35.6 | 66.7 | - |
| CORRECT | Acc@1 | 28.6 | **60.9** | **38.9** | **44.1** | **80.4** | **57.1** | **69.1** | **+20.1** |
| | Acc@3 | **50.0** | **94.2** | **88.9** | **88.2** | **88.2** | **87.8** | **88.2** | **+27.6** |
| | Acc@5 | **64.3** | **95.7** | **88.9** | **94.1** | **91.2** | **95.9** | **94.1** | **+28.7** |
| Synthesized by **GPT-5-Nano** | | | | | | | | | |
| Baseline | Acc@1 | 16.7 | 14.7 | 11.9 | 6.44 | 62.8 | 41.8 | 50.0 | - |
| | Acc@3 | 27.8 | 48.9 | 43.9 | 35.6 | 69.6 | 57.1 | 64.7 | - |
| | Acc@5 | 38.9 | 65.8 | 58.8 | 56.1 | 71.1 | 64.3 | 70.59 | - |
| CORRECT | Acc@1 | **30.6** | **35.8** | **32.7** | **16.9** | **80.4** | **57.1** | **69.1** | **+16.8** |
| | Acc@3 | **44.4** | **72.7** | **61.2** | **49.9** | **88.2** | **87.8** | **88.2** | **+20.1** |
| | Acc@5 | **52.8** | **84.3** | **77.2** | **69.0** | **91.2** | **95.9** | **94.1** | **+18.9** |

Table 2: Performance comparison across multiple datasets. All numbers are accuracy (%).

ances, performance gaps widen further: *CORRECT* reaches 94.2%, 91.2%, and 95.9% at Acc@5, compared to baseline scores of 63.8%, 78.0%, and 35.6%. *CORRECT* exhibits a similar trend in GPT-5-nano subset, with average improvements of +16.8%, +20.1%, and +18.9% across tolerance levels using Qwen-2.5-7b. Even on the challenging GAIA benchmark, *CORRECT* raises accuracy from 28.6% to 30.6%, while achieving near-perfect scores on several datasets at Acc@5.

**Strong Schema Transferability across Datasets and Models.** Figure 6 shows that schemas distilled from human-crafted trajectories transfer effectively to algorithm-generated data. Across GPT-5-nano, Gemini-2.5-Flash, and Qwen-7B, *CORRECT* with transferred schemas consistently outperforms baselines, with Gemini-2.5-Flash improving from 31.8% to 36.5%. This cross-domain transferability indicates that distilled schemas capture fundamental error patterns.

Moreover, Figure 7 demonstrates that *CORRECT* benefits directly from model upgrades: using GPT-5 instead of Qwen-72B as the schema generator raises detection accuracy from 8.6% to 12.2% for GPT-5-nano and from 10.3% to 12.2% for Qwen-2.5-7B. These adaptive gains confirm that better models yield higher-quality schemas that immediately enhance downstream performance. Together,

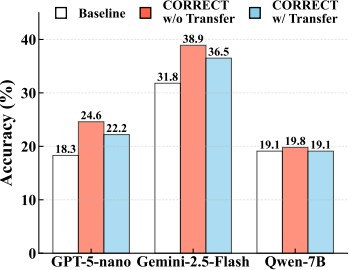 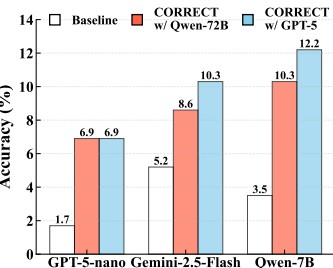 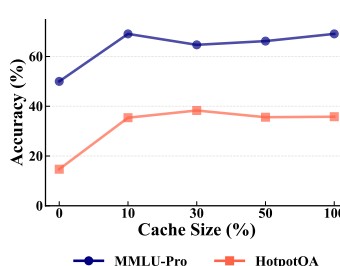

Figure 6: *CORRECT* delivers consistent improvements on Algorithm-Generated, with schemata generated and transferred on Hand-Crafted dataset.

Figure 7: *CORRECT* can adaptively upgrade its performance on the Hand-Crafted dataset with model upgrade.

Figure 8: Correct deliver robust improvements under different cache sizes.

these results establish *CORRECT* as a flexible, upgrade-compatible framework that leverages both existing schema libraries and future model advances without architectural changes.

## 5.2 ONLINE DEPLOYMENTS

Real deployments often face continuously evolving request distributions. To approximate this setting, we initialize CORRECT using schemata distilled from WHO&WHEN and gradually augment the cache with schemata derived from incoming HotpotQA and WikiMQA trajectories. As shown in Table 3, *CORRECT* maintains stable accuracy even when only 10–20% of the newly observed schemata are incorporated. This reflects both the strong transferability of our schemata and the small amount of on-task data required for CORRECT to adapt effectively in dynamic environments.

Table 3: Streaming-style adaptation: accuracy as more schemata from new domains are added.

| Dataset | 0–10% | 0–20% | 0–50% | 0–100% |
|---|---|---|---|---|
| Hotpot | 35.8 | 40.6 | 41.2 | 39.4 |
| WikiMQA | 70.8 | 68.6 | 67.6 | 69.1 |

## 5.3 ABLATION STUDIES

**Impact of Schema Repository Size.** Figure 8 shows *CORRECT*'s robustness to cold-start scenarios and varying cache sizes. On CORRECT-ERROR, even with only 10% of the schema library, *CORRECT* achieves 69.1% and 35.4% Acc@1 on MMLU_pro and HotpotQA, substantially outperforming baselines. Performance improves steadily with larger caches but plateaus beyond 50% (tens of schemas), suggesting that a relatively small set of diverse schemas captures most common error patterns. This logarithmic growth pattern validates our clustering-based extraction strategy: error patterns are finite and reusable across trajectories. Importantly, *CORRECT* retains ∼90% of peak performance with only 30% of the cache, highlighting efficiency for practical deployments.

**Impact of Number of Schemas in Online Error Recognition.** Figure 9 demonstrates that retrieving and using a single schema already greatly improves baseline Acc@1 (12.1% vs. 8.6%). Accuracy increases as more schemas are added (13.8% with 5 schemas, 15.5% with 10), though gains diminish (1→5 adds +1.7%; 5→10 adds only +1.7%). At higher tolerances (Acc@5), settings converge to ∼48%. These results show that a small set of well-matched schemas already efficiently captures most critical patterns, while additional schemas offer limited incremental value.

**Comparison with Oracle Error Schema.** To estimate the upper bound of our framework, we compare *CORRECT* against an oracle configuration where each trajectory uses its own ground-truth schema. As shown in Figure 10, the oracle achieves superior step-level accuracy, while *CORRECT* with 5 retrieved schemas reaches 71.5%—104% of oracle performance. This narrow gap shows that our semantic retrieval strategy effectively identifies schemas encoding near-equivalent knowledge to

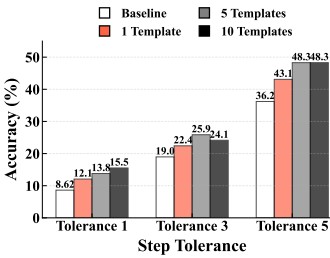
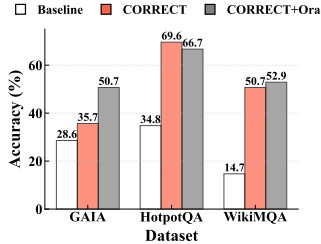
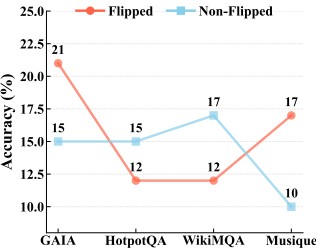

Figure 9: Performance of Correct with different number of error schemata on Handcrafted subset of Who&When.

Figure 10: Performance of Correct and Correct with the variation using oracle error schema on CORRECT-Error.

Figure 11: LLMs have low performance in recognizing the errors when they encounter them.

trajectory-specific patterns. The convergence toward oracle performance validates that diverse MAS error patterns often share structural regularities, which can be captured via a finite schema library.

**LLMs can't Recognize Errors During Execution** Figure 11 shows that LLMs have limited metacognitive ability to detect their own errors during execution. When asked to identify injected errors at the exact step, they achieve only 21% accuracy on *flipped* trajectories (where errors alter the final answer) and 17–18% on *non-flipped* trajectories (where errors do not affect the outcome). This reveals a fundamental limitation: *agents lack the self-awareness to recognize their own mistakes, regardless of downstream task success.* These findings underscore the necessity of external error-detection mechanisms like *CORRECT*, as relying on self-monitoring leaves MAS vulnerable.

**Overhead of Schema-Guided Recognition** CORRECT introduces only minimal overhead in practice. As shown in Table 4, API usage increases only slightly (0.86→0.89) and wall-clock latency rises from 44s to 47s on WHO&WHEN Hand-Crafted—an overhead of less than 5%. Since schema retrieval adds only a small amount of contextual guidance to the prompt and requires no fine-tuning, the approach remains lightweight and compatible with real-time MAS deployments.

**Sensitivity to Similarity and Hotness Thresholds.** CORRECT is robust to its two primary hyperparameters: the similarity threshold $\delta$ and the schema "hotness" threshold $\theta_{hot}$. As shown in Table A.15 and Table A.15, varying $\theta_{hot}$ (which governs how frequently high-access schemata are refined) produces only small but consistent gains on ARC (80.4→81.3 as $\theta_{hot}$ increases from 0 to 0.3), while adjusting $\delta$ mainly trades off cache size against accuracy (76.5, 78.1, and 78.9 for $\delta=0.6, 0.7, 0.8$ respectively). They indicate that $\delta=0.7$ and refining the top 20% most-accessed schemata provide a strong balance, and that CORRECT's performance is stable across settings.

**Robustness to Retrieval Noise.** To evaluate whether CORRECT depends on perfectly matched schemata, we inject irrelevant "hard-negative" schemata into the top-$k$ retrieval pool while keeping $k=5$ fixed. Accuracy decreases moderately as noise increases (e.g., Qwen-7B drops from 12.3% to 6.9% when all five retrieved schemata are random), yet CORRECT consistently outperforms the baseline (3.5%). GPT-5 shows a similar trend (17.2% → 10.7% vs. baseline 8.6%). These results show that CORRECT does not rely on perfect retrieval—schema guidance remains useful even when multiple partially mismatched schemata are included. Full results appear in Appendix A.16.

## 6 CONCLUSION

We introduced CORRECT, the first schema-guided framework that distills recurrent MAS failures into compact, reusable schemata, enabling accurate, lightweight, and training-free identification of decisive errors in new runs. Complementing this, we release CORRECT-ERROR, a large-scale, high-fidelity benchmark capturing realistic error patterns. Across diverse tasks, models, and deployment scenarios, CORRECT significantly outperforms existing advances, offering a practical, generalizable path toward reliable, interpretable, and scalable MAS deployment.

**Reproducibility Statement:** Due to space limits, we add the statement to Appendix A.1.

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

# A APPENDIX

## A.1 REPRODUCIBILITY STATEMENT

**Reproducibility Statement.** To ensure reproducibility of our work, we provide complete algorithmic details in Algorithm 1 and comprehensive explanation in §3. All experiments use publicly available datasets (WHO&WHEN and our *CORRECT*-Error benchmark, which is going to be released after acceptance) and publicly accessible models (Qwen, GPT, Gemini series models) as detailed in §5. The appendix contains extensive implementation details including: exact hyper-parameters and details for implementation (§A.3), complete prompts for error schema generation (§A.5), and prompts for online schema-guided detection (§A.6). We also include more details for creating the *CORRECT*-Error in (§A.4). We will release our code and the *CORRECT*-Error dataset upon publication to facilitate reproduction and future research.

## A.2 THE USE OF LARGE LANGUAGE MODELS (LLMS) STATEMENT

**Use of Large Language Models.** We used LLMs solely for editorial polishing of the manuscript text. LLMs played no role in research ideation.

## A.3 SPECIFICS OF EXPERIMENTAL SETTINGS

We implement our evaluation pipeline based on Zhang et al. (2025). We host open-source models using vLLM(Kwon et al., 2023) and access GPT-series models(Achiam et al., 2023) via the OpenAI API. To handle long contexts exceeding standard model limits for Qwen models(Yang et al., 2025), we employ RoPE(Su et al., 2024) scaling with 4× length extension using the ”yarn”(Peng et al., 2023b) scaling type. To simulate realistic deployment scenarios where ground truth is unknown, we exclude the correct answer from evaluation prompts.

For our method, we first generate all the error schemata using GPT-5 model. We then derive a similarity mapping to assign error schemata based on the semantic embedding decoded by BAAI-BPE-M3 model(Chen et al., 2024). As shown in Appendix A.12, *CORRECT* is robust to aggressive truncation: using 500, 1000, or 2000 characters per turn yields nearly identical performance, indicating that long MAS logs (often 60k+ tokens) do not pose a practical limitation for schema retrieval.

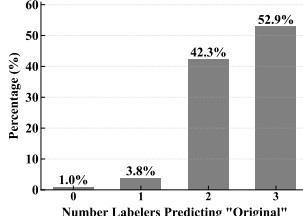

To avoid the data leakage, we mask each trajectory itself and avoid receiving its own error schema. We decide the number of error schemata from the experiments using Qwen-2.5-7b models on Hand-Crafted dataset, Algorithm-Generated dataset, and HotpotQA dataset of CORRECT-Error. We use the same number of error schemata across all models and all datasets in CORRECT-Error.

Figure 12: Percentage of human labelers to believe the trajectory is not synthesized.

Specifically, we use 1 error schema for all experiments on the Algorithm-Generated dataset, 10 error schemata for all experiments on the Hand-Crafted dataset, and 5 error schemata for all experiments on CORRECT-Error.

## A.4 MORE DETAILS OF THE CORRECT-ERROR

We implemented a variant of Magentic-One (Fourney et al., 2024) using selector group based workflow control using AutoGen (Wu et al., 2024) to generate CORRECT-Error.

Apart from the figures we showed in section 4, we observed strong inter-annotator consensus: 94.4% of synthetic trajectories fooled at least two labelers, while 52.9% were unanimously mistaken for genuine errors. We show the distribution in Figure 12

## A.5 PROMPTS FOR OFFLINE ERROR SCHEMA GENERATION

We show the prompts we used for offline schema generation in Fig A.5

**Error Schema Generation**

"""Given an error analysis from a multi-agent conversation, create an error schema to help identify similar errors in the future.
Context:
Question: {question}
Ground Truth: {ground_truth}
Error Agent: {mistake_agent}
Error Step: {mistake_step}
Error Reason: {mistake_reason}
Conversation History: {chat_content}
Based on this error case, please create a error schema that will help IDENTIFY similar errors in future conversations. Focus primarily on recognition patterns rather than mitigation strategies. The schema should include:
1. Error Signatures: - What distinctive patterns or signals indicate this type of error is occurring? - What are the telltale signs in the agent's behavior or responses?
2. Error Context Analysis: - What contextual conditions typically surround this type of error? - What sequence of interactions tends to precede this error?
3. Detection Heuristics: - What specific questions can be asked to determine if this error is present? - What analytical framework can help identify this error pattern? - What key phrases or conversation patterns serve as reliable indicators?
Please format your response as a structured schema that focuses specifically on ERROR IDENTIFICATION, not on how to improve agent behavior.
Provide a concise, actionable schema in the following format:
Agent Name: {mistake_agent}
Step Number: {mistake_step}
Reason for Mistake: [Your analysis of why this specific error occurred and how to identify similar patterns] """

**Schema-guided generation**

"HOW TO USE THIS REFERENCE EXAMPLE:"
"This template demonstrates one type of error pattern for reference. To apply it to your analysis:"
"1. Study the ERROR PATTERN shown: What type of mistake does this example identify?"
"2. Use this as reference to analyze YOUR conversation:"
" • Read through your conversation systematically (Step 0, Step 1, Step 2...)"
" • At each step, ask: 'Is there an error here, and does it match this pattern or a different one?'"
" • The error in your case may follow the same pattern or be completely different"
"3. Remember this is just a reference example:"
" • Your error may occur at any step number"
" • Your error may be a different type entirely"
" • Use this template to help you recognize what errors look like, not to assume your error matches"

## A.6 PROMPTS FOR ONLINE SCHEMA-GUIDED GENERATION

We show the prompts we used for online schema-guided generation in Fig A.6

## A.7 OVERHEAD ANALYSIS

## A.8 COMPARISON TO HUMAN-CURATED TAXONOMIES

We further compare and contrast the error schemata generated by CORRECT with human-curated taxonomies in MAST:

Table 4: Overhead of CORRECT on WHO&WHEN Hand-Crafted. API cost is normalized per query; time is end-to-end latency with Qwen-7B.

| Method | API Cost | Time |
|--------|----------|------|
| Baseline | 0.86 | 44s |
| CORRECT | 0.89 | 47s |

**Error Schemata vs. Static Human Taxonomies**    Human-curated taxonomies face several limitations:

- **Lack of granularity.** Human-defined categories (e.g., "wrong tool use," "missing precondition") operate at coarse conceptual levels and do not provide the fine-grained, step-specific patterns required for localizing errors within trajectories.
- **High manual cost and slow iteration.** Curating and maintaining a taxonomy requires extensive domain expertise. As models, tools, and task distributions shift, these taxonomies quickly become outdated.
- **Limited adaptability under distribution drift.** Static taxonomies cannot capture new, emerging failure modes as the environment or agents evolve.

**Advantages of CORRECT**    CORRECT automatically distills fine-grained, step-level schemata from observed trajectories and updates them asynchronously without adding online inference overhead. This enables CORRECT to track evolving failure modes and maintain accuracy under distribution drift—capabilities that static human-curated taxonomies cannot provide in large-scale deployments.

### A.9    HUMAN VERIFICATION, AUTOMATED VALIDATION, AND SCALABILITY

Human involvement during schema construction can introduce overhead, but CORRECT is not dependent on manual supervision. In practical deployments (e.g., conversational systems such as ChatGPT or Gemini), user feedback is already collected passively and can naturally serve as weak supervision without incurring additional annotation costs. Moreover, each schema is reused across many queries, so even a small number of validated schemata provides substantial amortized benefit: as shown in Figure 8, only 58 schemata are sufficient to improve HotpotQA error-detection accuracy from 14.7 to 35.4.

When human input is unavailable, CORRECT can instead employ LLM-as-a-judge with confidence filtering to automatically validate successful trajectories before inserting new schemata into the cache. This aligns with recent efforts such as EcoAssistant, which leverage human-verified or automatically filtered caches to improve long-horizon reasoning. Exploring more automated or semi-supervised schema-validation pipelines—while preserving interpretability and reliability—is a promising direction for future work.

### A.10    FINE-TUNING BASELINE DETAILS

This section provides full technical details for the supervised fine-tuning (SFT) baseline referenced in the main paper.

**Training Objective and Data Construction**    Following prior work (**?**Peng et al., 2023a; **?**), we train the model with a standard cross-entropy loss over the assistant responses. To avoid data leakage, SFT for the Hand-Crafted evaluation uses training data from the Algorithm-Generated split, and vice versa.

Each training instance is formatted as:

```
messages = [
{"role": "system", "content":  "You are an AI assistant
specialized in analyzing multi-agent conversations to identify
errors."},
```

```
{"role":  "user", "content":  prompt},
{"role":  "assistant", "content":  answer}
]
```

Only the assistant output is used as the training target.

**Hyperparameters**   We perform a grid search over learning rates {1e-6, 5e-6, 1e-5, 5e-5} and batch sizes {8, 16, 32}. The selected hyperparameters for the two SFT models are:

|  | Learning Rate | Batch Size |
|---|---|---|
| Hand-Crafted SFT | 5e-5 | 16 |
| Algorithm-Generated SFT | 1e-5 | 32 |

These models are trained on the full trajectory dataset for each split to encode domain-specific failure patterns.

### A.11   PROMPT FOR NATURAL ERROR INJECTION

### A.12   ROBUSTNESS TO CONTEXT-LENGTH TRUNCATION

We use the BGE-M3 encoder (context window 8192) to embed multi-agent trajectories. Because raw MAS logs can exceed 60k–70k tokens, we truncate each turn to a fixed character budget before embedding. Table 5 shows that CORRECT is highly robust to this truncation: using 500, 1000, or 2000 characters per turn yields nearly identical performance.

Table 5: Effect of per-turn truncation length on schema retrieval accuracy.

| Dataset | 500 chars | 1000 chars | 2000 chars |
|---|---|---|---|
| Hand-Crafted | 12.2 | 12.2 | 12.2 |
| Algorithm-Generated | 19.8 | 19.8 | 19.2 |

These results show that CORRECT's retrieval mechanism remains stable even under aggressive truncation, and long-context MAS logs do not hinder schema matching in practice.

### A.13   FAILURE CASES: MULTI-ERROR TRAJECTORIES

CORRECT may misfire when a trajectory contains multiple errors and a later, higher-salience failure dominates the schema match. Because CORRECT follows the decisiveness definition of (**?**), it is designed to identify the earliest decisive error; however, when downstream errors exhibit stronger structural signals, the retrieved schema may align more closely with these later steps.

**Example.**   In the example below, the human-labeled decisive error occurs at Step 4:

> *"WebSurfer failed to locate the specific volume in the University of Leicester paper due to incomplete data retrieval and insufficient PDF analysis."*

CORRECT instead predicts Step 12 as the decisive error:

> *"The agent remained on the search-results page instead of navigating into the DOI page containing the required endnote information."*

**Schema Match.**   The retrieved schema (abridged) emphasizes:

- **Error signature:** incomplete or insufficient search criteria,
- **Context:** tasks requiring precise filtering of external data sources,
- **Heuristic:** "Is the agent's search action complete and accurate?"

Both Step 4 and Step 12 satisfy these conditions, but Step 12 provides a more overt instance of incorrect search behavior, making it a stronger surface-level match for the schema.

```
You are analyzing an error pattern to create an injection
plan.

THOUGHT TEMPLATE (Error Detection Guidelines):
{thought_template}

ERROR INFORMATION:
- Agent: {mistake_agent}
- Step: {mistake_step}
- Reason: {mistake_reason}

ERROR TRAJECTORY:
{error_formatted}

CORRECT TRAJECTORY:
{correct_formatted}

CRITICAL REQUIREMENTS:
1.  Injection point must be semantically similar to the
original error.
2.  Error must flow naturally from that point.
3.  Consider cascade effects on all later steps.
4.  For consecutive CodeExecutor steps, modify only the final
step's output.

STEP-BY-STEP ANALYSIS REQUIRED:
Step 1:  Analyze the original error and identify what went
wrong.
Step 2:  Map to semantically similar steps in the correct
trajectory.
Step 3:  Select the proper injection point.
Step 4:  Describe how the error should be replicated.
Step 5:  Assess how the injected error propagates.

VERIFICATION CHECKLIST:
- Semantically similar injection point?
- Natural error flow?
- Cascade effects considered?
- Final CodeExecutor step modified if consecutive?
- Realistic and agent-consistent error?

Output format:
<injection_step>[...]</injection_step>
<injection_agent>[...]</injection_agent>
<error_pattern>[...]</error_pattern>
<injection_strategy>[...]</injection_strategy>
<expected_impact>[...]</expected_impact>
```

```
You are tasked with modifying a message to inject a specific error.  Follow the steps below
carefully.

ORIGINAL MESSAGE:
Agent: {agent}
Content: {original_content}

INJECTION PLAN:
Error Pattern: {injection_plan['error_pattern']}
Strategy: {injection_plan['error_strategy']}
Expected Impact: {injection_plan['expected_impact']}

INSTRUCTIONS: Complete each section below in order.  Show your
thinking process.

<STEP1_ANALYSIS>
- What is the agent's formatting style?
- What was the agent trying to communicate?
- What is the context of this message?
</STEP1_ANALYSIS>

<STEP2_ERROR_UNDERSTANDING>
- What specific error am I injecting? {injection_plan['error_strategy']}
- Why would this error naturally occur?
- How does this relate to the error pattern
{injection_plan['error_pattern']}?
</STEP2_ERROR_UNDERSTANDING>

<STEP3_MODIFICATION_PLAN>
- What exact change will I make?
- How will I maintain the agent's style?
- Why is this change realistic?
</STEP3_MODIFICATION_PLAN>

<STEP4_VERIFICATION>
☐ Maintains {agent}'s style and format?
☐ Implements strategy exactly?
☐ Believable to the agent?
☐ Causes {injection_plan['expected_impact']}?
☐ Within agent capabilities?
☐ Realistic error?
☐ Leads to incorrect final answer?
</STEP4_VERIFICATION>

<MODIFIED_CONTENT>
[Put ONLY the modified content here, exactly as the agent
would output it]
</MODIFIED_CONTENT>
```

**Discussion.** Such multi-error trajectories represent the primary category where CORRECT may deviate from human-annotated earliest-error labels. In practice, they are uncommon, but they highlight an inherent challenge of schema-guided detection when structurally similar failures occur at multiple points in a trajectory.

### A.14 TRANSFERABILITY TO THE ERROR-CATEGORY SETTING OF (?)

The dataset introduced in (Cemri et al., 2025) provides error *categories* but does not include step-level annotations. This differs from our formulation, where CORRECT predicts the earliest decisive step in a trajectory. Because of this structural mismatch, a direct step-level comparison is not possible. Nevertheless, to evaluate cross-formulation transferability, we adapt CORRECT to the category-labeling task by providing only the retrieved schemata and asking the model to output an error *type*.

Using Qwen-7B, CORRECT improves recall on both released datasets:

Table 6: CORRECT adapted to the error category detection setting of (Cemri et al., 2025).

| Dataset | Baseline | CORRECT |
|---|---|---|
| ProgramDev + ChatDev | 15.7 | 16.6 |
| AG2 + GSM8K | 11.9 | 15.8 |

These improvements show that CORRECT retains its benefits even when reduced to category-level supervision, demonstrating compatibility with the formulation used in (?).

### A.15 SENSITIVITY TO SIMILARITY AND HOTNESS THRESHOLD

Table 7: Sensitivity to the schema-refinement threshold $\theta_{\text{hot}}$.

| $\theta_{\text{hot}}$ | 0.0 | 0.1 | 0.2 | 0.3 |
|---|---|---|---|---|
| ARC Accuracy | 80.4 | 80.6 | 81.2 | 81.3 |

**Effect of $\delta$.** Lowering the similarity threshold reduces cache size but slightly harms accuracy, whereas $\delta = 0.7$ offers a strong balance:

Table 8: Sensitivity to the similarity threshold $\delta$.

| $\delta$ | 0.6 | 0.7 | 0.8 |
|---|---|---|---|
| ARC Accuracy | 76.5 | 78.1 | 78.9 |

### A.16 ROBUSTNESS TO RETRIEVAL NOISE

To assess CORRECT's resilience to mismatched schema retrieval, we inject random, irrelevant schemata ("hard negatives") into the retrieval pool while keeping the total number of retrieved schemata fixed at $k=5$. The tables below report performance as the number of injected random schemata increases.

Table 9: Impact of random (irrelevant) schemata injected into the retrieval pool (Qwen-7B).

| Model | Baseline | CORRECT | +1 Rand | +3 Rand | +4 Rand | +5 Rand |
|---|---|---|---|---|---|---|
| Qwen-7B | 3.5 | 12.3 | 10.7 | 10.7 | 10.7 | 6.9 |

Even with multiple hard negatives included, CORRECT consistently improves over the baseline, demonstrating strong robustness to retrieval noise and confirming that schema-guided reasoning does not depend on perfect semantic matching.

Table 10: Impact of retrieval noise on GPT-5.

| Model | Baseline | CORRECT | +1 Rand | +3 Rand | +4 Rand | +5 Rand |
|-------|----------|---------|---------|---------|---------|---------|
| GPT-5 | 8.6 | 17.2 | 13.8 | 15.5 | 13.8 | 12.1 |

