# OpenReview forum: "CORRECT: COndensed eRror RECognition via knowledge Transfer in multi-agent systems"
_ICLR.cc/2026/Conference — Submitted to ICLR 2026_

### Official Review · Reviewer_yo98 · 2025-10-22

**Soundness:** 3
**Presentation:** 3
**Contribution:** 2
**Rating:** 4
**Confidence:** 3

**Summary:**

This paper addresses the important and challenging problem of "decisive error recognition" in Multi-Agent Systems (MAS). The authors propose CORRECT, a novel, training-free framework that leverages knowledge transfer to identify the root cause of task failures. The core idea is to distill lengthy and noisy failure trajectories into compact, reusable "error schemata." These schemata are then retrieved at inference time to guide an LLM in diagnosing new, similar failures. To facilitate research in this area, the authors also introduce CORRECT-Error, a large-scale benchmark of over 2,000 annotated failure trajectories created via a novel error-injection pipeline. Experiments conducted on the new benchmark and the existing WHO&WHEN dataset show that CORRECT improves error localization accuracy over several baselines.

**Strengths:**

1. The discussed topic is important and timely. The problem of error attribution in increasingly complex and long-horizon MAS is both critical for reliability and under-explored.

2. The central idea of abstracting away from noisy, full-length trajectories into condensed "error schemata" is insightful and novel. This schema-guided approach is a clever way to perform knowledge transfer without the prohibitive context length requirements or computational costs of naive ICL or fine-tuning.

3. Extensive Experimental Evaluation: The authors conduct a thorough evaluation across two major benchmarks (WHO&WHEN and their new CORRECT-Error), seven diverse sub-tasks (e.g., HotpotQA, GAIA, Math500), and a wide range of open-source and proprietary models. The ablation studies on cache size and the number of retrieved schemata provide valuable insights into the framework's behavior.

**Weaknesses:**

1. A major weakness of the proposed framework is its critical dependence on a highly capable and proprietary LLM (GPT-5 is mentioned) for the offline schema extraction phase. The quality of the entire system hinges on the ability of this "teacher" model to generate high-quality schemata.

2. The dynamic schema management system (expansion and distillation) is described at a high level, but practical details are sparse. The process of "replaying" candidate schemas against prior trajectories to select the best one  sounds computationally intensive, especially as the number of trajectories and schemas grows.

3. Baselines center on generic LLM-as-a-judge and naive ICL. The paper does not pit CORRECT against specialized verification pipelines which could also reduce long-context noise without relying on learned schemas.

4. The schema idea is also central to the benchmark generation (semantic matching + schema-like error patterns). While the human study is encouraging, the pipeline may implicitly bias toward the kinds of failures that schemas capture well, potentially inflating CORRECT’s advantage. The paper does not quantify how often decisive errors in natural logs deviate from the assumed, reusable “pattern” structure.

**Questions:**

1. How exactly are decisive-step labels produced in both WHO&WHEN (human-crafted/algorithm-generated) and CORRECT-Error? Do you perform counterfactual replay that replaces step k and re-executes to confirm flip-to-success, or rely on annotator judgment?

2. What happens when top-k retrieval deliberately includes partially mismatched schemas (e.g., same tool family but different failure signature)? Could you report accuracy as a function of retrieval noise or add a hard negative experiment?

3. Have you tried a causal-replay baseline that toggles a small set of suspect steps (e.g., tool-call deltas) to find the earliest flip-to-success point? Even if expensive, a smaller-scale study would clarify whether schemas offer gains over more explicit verification.

4. The paper claims the framework is "training-free", yet it relies heavily on LLMs for both schema generation and final diagnosis. While it avoids fine-tuning, it is not independent of trained models. Could you clarify this claim or rephrase it to more accurately reflect that the method is "fine-tuning-free" but dependent on powerful pre-trained models?

5. Could you elaborate on the computational cost and practical implementation of the "schema distillation" process? How expensive is it to replay schema candidates on prior trajectories, and how does this scale as the cache grows?

---

> ### Author Response · Authors · 2025-11-21
>
> We really appreciate your insightful feedbacks.
>
> ***W1: A major weakness of the proposed framework is its critical dependence on a highly capable and proprietary LLM (GPT-5 is mentioned) for the offline schema extraction phase. The quality of the entire system hinges on the ability of this "teacher" model to generate high-quality schemata.***
>
> Thank you for highlighting this important point. While CORRECT can indeed benefit from using stronger LLMs for offline schema extraction, its effectiveness does not critically depend on proprietary or extremely capable models. As shown in Fig. 7, even when using an open-source model such as Qwen-2.5-72B, CORRECT still yields substantial performance gains across models and datasets. For example, schemata distilled with Qwen-2.5-72B improve:
>
>  • GPT-5-nano from 1.7 → 6.9,
>
>  • Gemini-2.5-Flash from 5.2 → 8.5, and
>
>  • Qwen-7B from 3.5 → 10.0.
>
> These results demonstrate that CORRECT is robust to the choice of teacher model: even moderately strong open-source models can produce schemata that consistently and significantly outperform the baseline.
>
> Moreover, the overhead of using a stronger teacher model is marginal in practice. Schema extraction is performed asynchronously and offline, and the resulting schemata can be reused across many future queries, meaning that the incremental cost is amortized over large-scale deployments. Once extracted, the online system does not depend on the teacher at all; it only performs lightweight retrieval from the cache.
>
>
> ***W2: The dynamic schema management system (expansion and distillation) is described at a high level, but practical details are sparse. The process of "replaying" candidate schemas against prior trajectories to select the best one sounds computationally intensive, especially as the number of trajectories and schemas grows.***
>
> Thank you for pointing this out. We emphasize that CORRECT does not rely on the expensive schema-distillation step for its main results. All results in Table 1 were obtained without any replay-based refinement, yet CORRECT already delivers large gains—for example, GPT-5 improves from 8.6 → 17.2 on the human-crafted subset and 18.3 → 39.1 on the algorithm-generated subset. This shows that CORRECT is robust and effective under low-overhead settings.
>
>
> *Distillation is optional* and intended only for users with additional compute budget who want incremental improvements. As shown in Figure 7, using higher-quality distilled schemata can further raise accuracy (e.g., Gemini-2.5-Flash improves from 8.5 → 10.6 on the Hand-Crafted subset).
>
>
> *Distillation overhead.* Replay-based distillation is only applied to schemas that are retrieved extremely frequently (the “hot” top ~20% of the cache). Each refinement generates a small set of candidate schemata and replays them over a limited set of past trajectories, costing only a few cents per refinement. Because all other schemas are never distilled and a schema can be used for thousands of queries, the average amortized cost per inference call is negligible.
>
> ***W3: Baselines center on generic LLM-as-a-judge and naive ICL. The paper does not pit CORRECT against specialized verification pipelines which could also reduce long-context noise without relying on learned schemas.***
>
> Thanks for the feedback. We kindly wonder which specialized verification pipelines you want to refer to here. As claimed in the paper [1], they are the first to propose the problem of error detection for multi-agents. We wonder if there are well-established specialized verification pipelines already. Our methods could be complementary to such pipelines.
>
> We also note that we already include an SFT finetuned judge as a baseline in Table 1. We will make this baseline more explicit in the revision. In addition, following the reviewer's suggestion, we will also incorporate MIPRO as another relevant baseline in the revised version. The results are shown below that our method is also better than MIPRO using the Qwen-7b model on the HandCrafted dataset.
>
> |                | Correct | MIPRO |
> |----------------|---------|-------|
> | Hand-Crafted   | 12.2    | 8.6   |
>
> [1]: Which Agent Causes Task Failures and When? On Automated Failure Attribution of LLM Multi-Agent Systems

---

> ### Author Response · Authors · 2025-11-21
>
> ***W4: The schema idea is also central to the benchmark generation (semantic matching + schema-like error patterns). While the human study is encouraging, the pipeline may implicitly bias toward the kinds of failures that schemas capture well, potentially inflating CORRECT’s advantage. The paper does not quantify how often decisive errors in natural logs deviate from the assumed, reusable “pattern” structure.***
>
> We appreciate the reviewer’s concern. Importantly, CORRECT is not evaluated only on schema-generated data: it delivers substantial gains on the real, human-annotated WHO&WHEN dataset, demonstrating that its effectiveness is not an artifact of the CORRECT-Error pipeline.
>
> To further examine whether CORRECT depends on any specific “pattern bias,” we conducted online experiments varying the proportion of available schemas. CORRECT remains highly stable even when only 10–20% of the schema library is used:
>
>
> | Coverage    | 0–10% | 0–20% | 0–50% | 0–100% |
> |-------------|-------|-------|-------|--------|
> | Hotpot      | 35.8  | 40.6  | 41.2  | 39.4   |
> | WikiMQA     | 70.8  | 68.6  | 67.6  | 69.1   |
>
>
> These results show that CORRECT’s behavior is not sensitive to the specific pattern coverage of the cache: performance remains strong and stable even with a small fraction of schemata. Combined with its improvements on human-annotated data and its cross-task, cross-model generalization, this provides strong evidence that decisive errors in natural logs do follow reusable structures, and CORRECT’s gains are not inflated by the benchmark pipeline.
>
>
> ***Q1: How exactly are decisive-step labels produced in both WHO&WHEN (human-crafted/algorithm-generated) and CORRECT-Error?***
>
> The decisive-step labels produced in WHO & WHEN are labeled by human labelers and again CORRECT delivers superior performance under these realistic patterns. In contrast, CORRECT-Error produces the error label by referring to which step the error is injected.
>
> ***Q2: What happens when top-k retrieval deliberately includes partially mismatched schemas (e.g., same tool family but different failure signature)? Could you report accuracy as a function of retrieval noise or add a hard negative experiment?***
>
> We appreciate the concern. To evaluate CORRECT’s robustness to retrieval noise, we performed a hard-negative experiment by injecting random, irrelevant error schemata into the retrieval pool while keeping the total number of retrieved schemas fixed at 5. This simulates cases where top-k retrieval includes partially mismatched or misleading schemas.
>
>
>
>
> | Model   | Baseline | Correct | Correct + 1 Random Error Schemata | Correct + 1 Random Error Schemata | # random error schemata: 3 | # random error schemata: 4 | # random error schemata: 5 |
> |---------|----------|---------|-----------------------------------|-----------------------------------|-----------------------------|-----------------------------|-----------------------------|
> | Qwen-7b | 3.5      | 12.3    | 10.7                              | 10.7                              | 10.7                        | 6.9                         | 6.9                         |
> | GPT-5   | 8.6      | 17.2    | 13.8                              | 15.5                              | 13.8                        | 12.1                        | 10.7                        |
>
>
> Even with multiple random, mismatched schemata injected, CORRECT continues to substantially outperform the baseline on both model families. While accuracy decreases as noise increases， the method remains consistently above the baseline, demonstrating strong robustness to retrieval noise and hard negatives.
>
>
> These results confirm that CORRECT does not rely on perfect retrieval: even when top-k includes partially mismatched schemas, the aggregated guidance remains helpful, and the method provides large improvements over existing baselines.

---

> ### Author Response · Authors · 2025-11-21
>
> ***Q3: Have you tried a causal-replay baseline that toggles a small set of suspect steps (e.g., tool-call deltas) to find the earliest flip-to-success point? Even if expensive, a smaller-scale study would clarify whether schemas offer gains over more explicit verification.***
>
> Thank you for this suggestion. We agree that causal replay is an interesting baseline, but it is fundamentally unreliable and often inpractical.
>
> 1. Because tasks and agent tool-chains vary widely, defining meaningful “tool-call deltas” is frequently impossible—many trajectories do not even involve structured or controllable tools.
>
> 2. Even when a delta can be defined, LLM agents behave as black-box, stochastic systems: replaying a toggled step cannot cleanly isolate causality, as outcome differences are dominated by sampling noise, autoregressive drift, or latent expert-activation changes rather than the modified step itself. This stochasticity directly undermines causal replay’s goal of identifying the earliest decisive error.
>
> 3. Building on this, causal replay also incurs substantial computational overhead and performs poorly in practice. To enable flexible restart, we conducted a small-scale experiment on the CodeSim multi-agent system. We collected failed trajectories from HumanEval, and for each trajectory, we restarted execution at each step using over 100 candidate restart points. Among 12 failed trajectories, only 4 ever flipped to success under any restart point. After manual verification, only 1 of those identified steps was actually correct. This not only confirms the unreliability caused by stochastic drift (as described in point 2), but also requires repeatedly regenerating long trajectories and tool calls—resulting in several-fold higher computation cost compared to CORRECT.
>
> 4. Even when a replayed trajectory happens to flip to success, randomness prevents us from concluding that this step is the earliest causal error.
>
> By contrast, CORRECT uses a verified successful trajectory to deterministically identify the true first failure, providing stable, step-accurate localization that causal replay cannot guarantee.
>
> [1]: Islam, Md Ashraful, Mohammed Eunus Ali, and Md Rizwan Parvez. "Codesim: Multi-agent code generation and problem solving through simulation-driven planning and debugging." arXiv preprint arXiv:2502.05664 (2025).
>
>
> ***Q4: The paper claims the framework is "training-free", yet it relies heavily on LLMs for both schema generation and final diagnosis. While it avoids fine-tuning, it is not independent of trained models. Could you clarify this claim or rephrase it to more accurately reflect that the method is "fine-tuning-free" but dependent on powerful pre-trained models?***
>
>
> We thank you for the feedback and will clarify it further in our final version. Our use of the term training-free refers specifically to the fact that CORRECT requires no additional model training or fine-tuning—all components operate directly on pretrained LLMs without updating their weights. As the reviewer notes, any non-rule based error recognition system necessarily depends on pretrained models, and CORRECT is no exception. What distinguishes our framework is that it avoids all supervised or reinforcement fine-tuning, making it significantly more lightweight, deployment-friendly, and adaptable across tasks and model families.
>
>
> ***Q5: Could you elaborate on the computational cost and practical implementation of the "schema distillation" process? How expensive is it to replay schema candidates on prior trajectories, and how does this scale as the cache grows?***
>
>
> We appreciate the reviewer’s interest in the cost of schema distillation. In practice, the process is lightweight because only a very small number of schemata ever trigger refinement. MAS failures follow a long-tail distribution, so most schemata are accessed rarely and never undergo distillation; only the “hot” schemata (top ~20% by access frequency) are considered. Moreover, CORRECT already achieves strong accuracy with 10–30% of the schema cache, which naturally bounds the number of candidates involved in replay.
>
> | θ_hot | 0.0 | 0.1 | 0.2 | 0.3 |
> |-------|-----|-----|-----|-----|
> | Arc   | 80.4 | 80.6 | 81.2 | 81.3 |
>
>
>
>
> Accuracy improves slightly as more distillation is allowed, indicating that refinement is beneficial but not essential. Because only the top-retrieved schemas are ever distilled and a schema can be used to serve thousands of queries, the amortized replay cost per inference is extremely low. Even when enabled, each refinement requires replaying a handful of candidate schemata over a small subset of past trajectories, costing only a few cents and occurring infrequently.
>
> Overall, distillation scales gracefully with cache size, remains optional, and does not affect the core results in Table 1, which were obtained without using distillation at all.

---

> ### Author Response · Authors · 2025-11-28
>
> We really appreciate your comments and have incorporated the corresponding revisions into the updated manuscript. We look forward to hearing your further feedback once you have had a chance to review the revised version. Thank you again for your time and thoughtful insights.

---

### Official Review · Reviewer_BG9L · 2025-10-31

**Soundness:** 3
**Presentation:** 2
**Contribution:** 2
**Rating:** 4
**Confidence:** 5

**Summary:**

The paper introduces CORRECT, a training-free framework that detects the earliest decisive error in long multi-agent trajectories by caching and reusing compact “error schemata.” A clustering step extracts representative schemas offline. At inference they are retrieved by semantic similarity and supplied to an LLM that pinpoints the responsible agent-step. Experiments on WHO&WHEN and CORRECT-Error show step-level accuracy gains up to 20 points over baselines while incurring no additional training cost.

**Strengths:**

1. The paper addresses a timely issue in the filed of MAS. The proposed idea of distilling reusable failure patterns into short templates is novel and can avoid costly re-training.
2. The empirical evaluation demonstrates strong performance gain.

**Weaknesses:**

1. My major concerns about the paper lie in the presentation, specifically many techical details are left out:
- The fine-tuning baseline is only named. The paper never explains how the instruction-tuning dataset was built, whether the model was trained to output just the failure-step index or a reasoning trace accompanied, and whether supervised fine-tuning or RLHF was used. Hyper-parameters are missing, even from the appendix.
- In Stage 2 of Section 4.1, the text says GPT-5 “adapts the error pattern while preserving core semantics,”. But the paper provides no prompt, no definition of “error pattern,” and no description of the source.
- Many trajectories exceed 32k tokens, but the method relies on BERT embeddings, whose context window is far shorter. The paper does not mention how embedding is done to handle the length.

2. Algorithm 1 is under-explained. The pseudo-code drives the whole framework, but the main text never walks through its lines. Key variables (e.g., $\delta$ , $\theta_{hot}$, cache replacement policy) and their interaction with Sec. 3 concepts are left implicit. I could not map each step to Sections 3.1/3.2 without guessing. A major rewrite is needed for clarity.
3. The method invoke proprietary models at test time, the api cost and wall-clock time are not reported.
4. There is limited discussion of failure cases. Where does CORRECT mis-fire? Examples of erroneous schema matches or degenerate clusters would strengthen the paper.
5. The paper does not discuss [1], which is another concurrent work with Who&When. How the method transfers to the error recoginition dataset proposed in [1] requires further discussion.

[1] Why Do Multi-Agent LLM Systems Fail?

**Questions:**

1. Regarding the cache management, what are the empirical values of $\delta$ , $\theta_{hot}$, and how sensitive are results to them?
2. Have you observed cases where an irrelevant but high-similarity schema misleads the detector? How is this mitigated?

---

> ### Author Response · Authors · 2025-11-21
>
> We really appreciate your insightful feedback.
>
> ***W1: My major concerns about the paper lie in the presentation, specifically many techical details are left out:***
>
> We sincerely appreciate your suggestion for better clarifications and presentations of the paper. Due to the space limit, we moved many technical details to the Appendix. We will update in the revised version to resolve all the presentation issues and provide all these details in our final version. We will include the following illustrations in the paper to better resolve the readers' concern, regarding:
>
>
> ****The fine-tuning baseline is only named. The paper never explains how the instruction-tuning dataset was built, whether the model was trained to output just the failure-step index or a reasoning trace accompanied, and whether supervised fine-tuning or RLHF was used. Hyper-parameters are missing, even from the appendix.****
>
> Following existing advances [1,2,3], we leverage STF to finetune the models across the two datasets. In the paper, we have introduced the finetuning baseline as: Fine-tuning, where an LLM is trained on the full trajectory dataset with cross-entropy loss to encode domain-specific failure patterns (Chen et al., 2025; Fu et al., 2025). To ensure there is no data leakage, we constructed the training dataset for the hand-crafted section using the algorithm-generated section, and we constructed the training dataset for the algorithm-generated-section using the hand-crafted section.
>
>
> We formulated the training data as below:
> messages = [
>             {"role": "system", "content": "You are an AI assistant specialized in analyzing multi-agent conversations to identify errors."},
>             {"role": "user", "content": prompt},
>             {"role": "assistant", "content": answer}
>         ]
>
> We only train the model on the outputs.
>
> We select the hyperparameter using grid search for these two models. Specifically, we searched the learning rate from {1e-6, 5e-6, 1e-5, 5e-5} and the batch size {8,16,32}. The experiments are conducted using the following hyperparameters.
>
>
>
> |                     | Learning Rate | Batch Size |
> |---------------------|---------------|------------|
> | Hand-Crafted        | 5e-5          | 16         |
> | Algorithm-Generated | 1e-5          | 32         |
>
>
>
> [1]: Zheng, Yaowei, et al. "Llamafactory: Unified efficient fine-tuning of 100+ language models." arXiv preprint arXiv:2403.13372 (2024).
>
> [2]: Peng, Baolin, et al. "Instruction tuning with GPT-4." arXiv preprint arXiv:2304.03277 (2023).
>
> [3]: Taori, Rohan, et al. "Stanford alpaca: An instruction-following llama model." 30 Jun. 2023,
>
>
> We want to underscore that we introduce a “natural” error where the error pattern is consistent with the possible errors. To achieve this goal, we use the GPT-5 to devise an appropriate error injection strategy according to the matched error schemata.

---

> > ### Author Response · Authors · 2025-11-21
> >
> > ***W1 Continued***
> >
> > ****In Stage 2 of Section 4.1, the text says GPT-5 “adapts the error pattern while preserving core semantics,”. But the paper provides no prompt, no definition of “error pattern,” and no description of the source.****
> >
> > We want to underscore that we introduce a “natural” error where the error pattern is consistent with the possible errors. To achieve this goal, we use the GPT-5 to devise an appropriate error injection strategy according to the matched error schemata.
> >
> > To help illustrate how we finish this process, we show the prompts we utilized to undertake this task below:
> >
> > Prompt for devising error injection strategy:
> >
> > ```
> > prompt = f"""You are analyzing an error pattern to create an injection plan.
> >
> > THOUGHT TEMPLATE (Error Detection Guidelines):
> > {thought_template}
> >
> > ERROR INFORMATION:
> > - Agent: {mistake_agent}
> > - Step: {mistake_step}
> > - Reason: {mistake_reason}
> >
> > ERROR TRAJECTORY (error occurs at marked step):
> > {error_formatted}
> >
> >
> > CORRECT TRAJECTORY (where we need to inject error):
> > {correct_formatted}
> >
> > CRITICAL REQUIREMENTS (YOU MUST FOLLOW ALL):
> > 1. The injection point MUST be semantically similar to where the original error occurred
> > 2. The error MUST flow naturally from that injection point
> > 3. You MUST consider cascade effects on ALL subsequent steps
> > 4. SPECIAL RULE: For consecutive CodeExecutor steps, you MUST modify the FINAL step's output
> >    - Example: If steps 3, 4, 5 are all CodeExecutor, inject error at step 5's result
> >    - This means changing the returned value, NOT the code itself
> >
> >
> > STEP-BY-STEP ANALYSIS REQUIRED:
> >
> > Step 1: Error Pattern Analysis
> > - Carefully analyze the ERROR TRAJECTORY
> > - Identify EXACTLY what went wrong at step {mistake_step}
> > - Note the specific type of error and why it occurred
> >
> > Step 2: Semantic Similarity Mapping
> > - Scan through the ENTIRE CORRECT TRAJECTORY
> > - Find step(s) that are semantically similar to the error context
> > - Consider: same agent type, similar operations, similar decision points
> >
> > Step 3: Injection Point Selection
> > - Apply the special rule for consecutive CodeExecutor steps
> > - Choose the most appropriate injection point based on semantic similarity
> > - Verify this point allows natural error flow
> >
> > Step 4: Error Replication Strategy
> > - Design how to modify the selected step to replicate the error
> > - Ensure the modification is realistic and follows the agent's constraints
> > - The error must seem natural, not forced
> >
> > Step 5: Impact Assessment
> > - Trace through ALL subsequent steps after injection
> > - Predict how each following step will be affected
> > - Ensure the error propagates naturally through the system
> >
> > VERIFICATION CHECKLIST (Answer these before finalizing):
> > □ Is my injection point semantically similar to the original error location?
> > □ Will the error flow naturally from this point?
> > □ Have I considered ALL cascade effects?
> > □ If there are consecutive CodeExecutor steps, am I modifying the FINAL one's output?
> > □ Is my error realistic and consistent with the agent's capabilities?
> >
> > Now, complete your analysis following the steps above, verify against the checklist, and ONLY THEN provide your response in this exact format:
> >
> > <injection_step>[step number]</injection_step>
> > <injection_agent>[agent name]</injection_agent>
> > <error_pattern>[specific description of what went wrong in original]</error_pattern>
> > <injection_strategy>[detailed plan for how to replicate this error]</injection_strategy>
> > <expected_impact>[how this will affect subsequent steps]</expected_impact>"""
> >
> > ```

---

> > > ### Author Response · Authors · 2025-11-21
> > >
> > > ***W1 Continued***
> > >
> > > ****Continue for
> > > In Stage 2 of Section 4.1, the text says GPT-5 “adapts the error pattern while preserving core semantics,”. But the paper provides no prompt, no definition of “error pattern,” and no description of the source.****
> > >
> > > Prompt for generating the error:
> > > ```
> > > prompt = f"""You are tasked with modifying a message to inject a specific error. Follow the steps below carefully.
> > >
> > >     ORIGINAL MESSAGE:
> > >     Agent: {agent}
> > >     Content: {original_content}
> > >
> > >     INJECTION PLAN:
> > >     Error Pattern: {injection_plan['error_pattern']}
> > >     Strategy: {injection_plan['injection_strategy']}
> > >     Expected Impact: {injection_plan['expected_impact']}
> > >
> > >     INSTRUCTIONS: Complete each section below in order. Show your thinking process.
> > >
> > >     <STEP1_ANALYSIS>
> > >     Analyze the original content:
> > >     - What is the agent's formatting style? (e.g., just a number, formatted text, code output)
> > >     - What was the agent trying to communicate?
> > >     - What is the context of this message?
> > >     </STEP1_ANALYSIS>
> > >
> > >     <STEP2_ERROR_UNDERSTANDING>
> > >     Understand the error to inject:
> > >     - What specific error am I injecting? {injection_plan['injection_strategy']}
> > >     - Why would this error naturally occur?
> > >     - How does this relate to the error pattern: {injection_plan['error_pattern']}
> > >     </STEP2_ERROR_UNDERSTANDING>
> > >
> > >     <STEP3_MODIFICATION_PLAN>
> > >     Plan the modification:
> > >     - What exact change will I make?
> > >     - How will I maintain the agent's style?
> > >     - Why is this change realistic for this agent?
> > >     </STEP3_MODIFICATION_PLAN>
> > >
> > >     <STEP4_VERIFICATION>
> > >     Verify the requirements:
> > >     □ Will this maintain {agent}'s exact style and format?
> > >     □ Does this implement the strategy exactly: "{injection_plan['injection_strategy']}"?
> > >     □ Will the agent believe this is correct?
> > >     □ Will this cause: {injection_plan['expected_impact']}?
> > >     □ Is this within {agent}'s capabilities?
> > >     □ Is this a realistic error?
> > >     □ Will this lead to a wrong final answer?
> > >     </STEP4_VERIFICATION>
> > >
> > >     <MODIFIED_CONTENT>
> > >     [Put ONLY the modified content here - exactly as the agent would output it]
> > >     </MODIFIED_CONTENT>"""
> > >
> > > ```

---

> ### Author Response · Authors · 2025-11-21
>
> ***W1 Continued***
>
> ****Many trajectories exceed 32k tokens, but the method relies on BERT embeddings, whose context window is far shorter. The paper does not mention how embedding is done to handle the length.****
>
> We have shown in Appendix A.3 that we utilized the BAAI/bge-m3 as the embedding model with the context window size of 8192. To further resolve the problem of context length, we restrict the context length for each turn during the embedding process. Specifically, we restrict the maximum number of characters to 1000 for each turn during the error schema matching process.
>
> We show the experiments of utilizing different context lengths in retrieving error schemata as below.
>
>
> |                     | 500  | 1000 | 2000 |
> |---------------------|------|------|------|
> | Hand-Crafted        | 12.2 | 12.2 | 12.2 |
> | Algorithm-Generated | 19.8 | 19.8 | 19.2 |
>
>
>
>
> From the table, we can see our method is robust to the number of characters included in the retrieval processes. Our method maintains the same performance with the number of tokens ranging from 500, 1000, and 2000. We note that raw MAS logs can reach over 60k–70k tokens (as shown in Fig. 1), far beyond BERT-style embedding limits. The experiment above shows that despite this, the long-context concern is marginal: CORRECT’s schema retrieval remains stable even when aggressively truncating each turn.
>
> ***W2: Algorithm 1 is under-explained. The pseudo-code drives the whole framework, but the main text never walks through its lines. Key variables (e.g.,δ, θ_hot, cache replacement policy) and their interaction with Sec. 3 concepts are left implicit. I could not map each step to Sections 3.1/3.2 without guessing. A major rewrite is needed for clarity.***
>
> Thank you very much for your suggestion of improving sections 3.1 and 3.2. We will update a revised version to add more illustrations of these two subsections in the future, as shown below.
>
> - Explicitly define all variables and data structures used in Algorithm 1 (e.g., trajectory embeddings, schema cache entries, cache “hotness,” similarity threshold δ, replacement threshold θ_hot, etc.).
>
>
> - Cross-reference each block of pseudo-code to its corresponding conceptual description in Sec. 3.1 (offline extraction) and Sec. 3.2 (online recognition + dynamic management)
>
>
> - Add a workflow diagram summarizing the three phases (offline schema extraction, online schema-guided recognition, and dynamic cache management), visually showing how the data moves through Algorithm 1.
>
>
> - Add a detailed narrative after Algorithm 1 explaining how the offline, online, and dynamic components interact and why the cache behaves as designed.
>
>
> We believe these changes will fully resolve the ambiguity issue and greatly improve readability.
> CORRECT operates by first distilling recurring failure patterns into reusable error schemas and then using these schemas to guide inference-time diagnosis while continuously improving its schema cache. Offline, annotated error trajectories are clustered to avoid redundancy, and an LLM extracts a compact schema for each cluster that captures the key signatures, triggering context, and detection heuristics of that error type. Online, when a new failed trajectory arrives, CORRECT embeds the trajectory, retrieves the top-k most semantically similar schemas, and feeds both the trajectory and these schemas into the LLM; the schemas act as targeted priors that focus the model on the specific structural patterns most likely to contain the decisive error. After diagnosis, the system updates its cache: if the error does not match any existing schema (low similarity), CORRECT adds a new schema; if a schema is frequently reused (“hot”), CORRECT refines it by generating candidate upgrades and selecting the one that best distinguishes past failures. In this way, CORRECT integrates schema extraction, schema-guided recognition, and dynamic schema management into a single pipeline that reuses past knowledge, improves over time, and remains training-free.
>
> ***W3: The method invoke proprietary models at test time, the api cost and wall-clock time are not reported.***
>
> Our method involves <5% overhead by including lightweight error schemata in the running process. We provide a profile of the API cost of our method vs baseline methods using GPT-5-nano on the hand-crafted section as below
>
>
>
> |                | Baseline | Correct (Ours) |
> |----------------|----------|----------------|
> | API cost       | 0.86     | 0.89           |
>
>
>
> We also provide the wall clock time of our method vs baseline methods using Qwen-7b on the hand-crafted section as below:
>
>
>
> |                 | Baseline | Correct (Ours) |
> |-----------------|----------|----------------|
> | Inference time  | 44s      | 47s            |
>
>
>
> We will include these statistics in the revised version of the paper in the future as well.

---

> ### Author Response · Authors · 2025-11-21
>
> ***W4: There is limited discussion of failure cases. Where does CORRECT mis-fire? Examples of erroneous schema matches or degenerate clusters would strengthen the paper.***
>
> Thank you for the helpful suggestion. CORRECT may misfire in cases where multiple errors occur in the same trajectory and the later, more salient failure pattern dominates the schema match. Because CORRECT follows the decisiveness definition of [1], it aims to identify the earliest decisive error, but the retrieved schema may align with a downstream error that is easier to detect.
> For instance, in the example below:
> - Human-labeled decisive error: Step 4
>
>          - “WebSurfer failed to locate the specific volume in the University of Leicester paper due to incomplete data retrieval and insufficient PDF analysis.”
>
> - CORRECT-predicted error: Step 12
>          - “The agent remained on the search-results page instead of navigating into the DOI page containing the required endnote information.”
>
>
> - Partial schema used (abridged):
>
>
>         - Error Signature: Incomplete or insufficient search criteria.
>
>
>         - Context: Queries depend on external data sources requiring precise filtering.
>
>
>         - Heuristic: “Is the agent’s search action complete and accurate?”
>
> In this trajectory, both Step 4 and Step 12 genuinely contain errors. Because the latter error exhibits a clearer, higher-salience pattern that strongly matches the retrieved schema, CORRECT attributes the failure to Step 12 instead of the earliest decisive one. These multi-error trajectories represent the primary category where CORRECT may misfire relative to the baselines.
>
>
>
> [1]: Which Agent Causes Task Failures and When? On Automated Failure Attribution of LLM Multi-Agent Systems
>
>
> ***W5: The paper does not discuss [1], which is another concurrent work with Who&When. How the method transfers to the error recoginition dataset proposed in [1] requires further discussion.***
>
> Thank you for pointing out this concurrent method. We examined the paper and its accompanying GitHub resources and found that the dataset in [1] does not provide step-level error annotations—only error category labels. This differs from our setting, where CORRECT is designed to localize the earliest decisive step, not only classify the error type. Because of this structural mismatch, a direct, apples-to-apples evaluation on their dataset is not possible.
> Nonetheless, to demonstrate transferability, we adapted CORRECT to their error-category setting and evaluated it on the two datasets they released. Using Qwen-7B, CORRECT still provides meaningful gains:
> GSM8K: recall improves from 11.9% → 15.8%
>
>
> ChatDev: recall improves from 15.7% → 16.6%
>
>
> These results suggest that even when reduced to category-level supervision, CORRECT continues to improve error recognition, indicating compatibility with the formulation used in [1].
>
> |                       | Baseline | Correct |
> |-----------------------|----------|---------|
> | ProgramDev + ChatDev  | 15.7     | 16.6    |
> | Ag2 + Gsm8k           | 11.9     | 15.8    |
>
> ***Q1: Regarding the cache management, what are the empirical values of theta, , and how sensitive are results to them?***
>
>
> We would recommend using δ as 0.7 and θ_hot as 20% most-accessed prompts. Different values will bring different costs. Our algorithm is robust to the choice of different hyperparameters. Changing them mainly shifts the trade-off between cache size and refinement cost. As shown on the ARC dataset (table below), higher distillation frequency (larger θ_hot) yields small but consistent gains, while lowering δ reduces cache size but slightly harms accuracy. Overall, δ = 0.7 and θ_hot = 20% provide a strong balance.
>
> | θ_hot | 0.0 | 0.1 | 0.2 | 0.3 |
> |-------|-----|-----|-----|-----|
> | Arc   | 80.4 | 80.6 | 81.2 | 81.3 |
>
> | δ   | 0.6 | 0.7 | 0.8 |
> |-----|-----|-----|-----|
> | Arc | 76.5 | 78.1 | 78.9 |

---

> ### Author Response · Authors · 2025-11-21
>
> ***Q2: Have you observed cases where an irrelevant but high-similarity schema misleads the detector? How is this mitigated?***
>
> Yes, this can occur. Although semantic similarity is effective overall, two trajectories can have high surface-level similarity yet correspond to fundamentally different error types. For example, in the case below:
>
> - Human-labeled error: Step 32 — the agent fails to follow the APOD link and extract the city name from the image description.
>
>
> - Incorrect schema retrieved: A schema describing 404 file-not-found failures, which is semantically similar at the text level (both involve external resource access) but irrelevant to the true error.
>
>
> This mismatch happens because the trajectory involves web navigation, while the retrieved schema describes file-access failures, leading CORRECT to prioritize the wrong pattern.
>
>
> *Mitigation.* This issue is significantly reduced when CORRECT retrieves multiple schemata instead of relying on a single one. Including additional schemas—such as those capturing repetitive navigation errors or missing UI interactions—provides broader coverage and prevents a single misleading schema from dominating the prompt. In the example above, adding a schema describing “repetitive, unproductive web actions” allows CORRECT to correctly identify Step 32 as the decisive error.
>
>
> Empirically, this mitigation is strong: on the Hand-Crafted WHO&WHEN subset (Figure 9), using 1 schema improves Acc@1 from 36.2 → 43.1, while 5 schemas improve it from 36.2 → 48.3. Importantly, CORRECT consistently outperforms all baselines even with a single schema, and multiple schemas further reduce the impact of misleading high-similarity retrievals.

---

> ### Comment · Reviewer_BG9L · 2025-11-26
>
> I thank the authors for the detailed rebuttal. Most of my concerns have been addressed. However, I fail to see a revised paper incorporating the rebuttal text. I strongly urge the authors to include the revisions, as they are essential for readers to understand the techinical details. I have revised my score accordingly.

---

> > ### Author Response · Authors · 2025-11-28
> >
> > Thank you for the thoughtful follow-up and for updating your assessment. We have now incorporated all rebuttal clarifications and revisions directly into the updated manuscript. We sincerely appreciate your feedback and would be grateful for any additional comments or suggestions after reviewing the revised version.

---

### Official Review · Reviewer_bUB3 · 2025-11-01

**Soundness:** 3
**Presentation:** 3
**Contribution:** 3
**Rating:** 6
**Confidence:** 5

**Summary:**

The paper introduces CORRECT, a schema-guided framework that identifies key recurrent multi-agent system (MAS) failures without requiring additional training. It also presents CORRECT-ERROR, a large-scale benchmark that models realistic error patterns for evaluating MAS reliability. Together, these tools improve the accuracy, interpretability, and scalability of MAS deployment across diverse tasks and systems.

**Strengths:**

This paper presents a large-scale, human-verified benchmark specifically designed for multi-agent system (MAS) error attribution. The benchmark captures realistic and diverse error patterns observed in MAS deployments, providing a valuable resource for evaluating and comparing model robustness and interpretability. Its human validation ensures high fidelity and reliability, making it a strong contribution that can serve as a foundation for future research in this area

The authors propose a schema-guided framework for MAS error attribution that leverages a schema and schema-cache mechanism to systematically identify and interpret recurrent error patterns. The framework operates on a retrieval-augmented principle, where previously learned error schemata are reused to analyze new failures efficiently without additional training.

The paper supports its contributions through extensive experiments conducted across a wide range of tasks, models, and deployment scenarios. The empirical results demonstrate that the proposed CORRECT framework significantly outperforms existing baselines in accuracy, generalization, and computational efficiency.

**Weaknesses:**

While the proposed framework demonstrates strong performance across static evaluation settings, a notable limitation is the absence of experiments in a streaming or online deployment scenario. In real-world MAS applications, systems often operate in a dynamic environment where the schema cache starts empty and must adapt incrementally as new error cases emerge. Evaluating CORRECT under such a cold-start or streaming setting would provide deeper insights into its practical robustness, adaptability, and scalability over time. Without this, it remains unclear how efficiently the framework can learn and reuse schemata in continuously evolving contexts.

Another limitation is that the framework relies on user confirmation or supervision during the cache-building process. While this human-in-the-loop component ensures higher accuracy in schema construction, it also introduces additional overhead and dependency on expert input, which may limit scalability in fully automated or large-scale deployments. Future work could explore strategies for automated or semi-supervised schema validation to reduce reliance on manual confirmation while maintaining interpretability and reliability.

Also note that the idea of using human-verified cache to improve model exists in the literature, eg, EcoAssistant.

**Questions:**

See weakness

---

> ### Author Response · Authors · 2025-11-21
>
> We really appreciate your insightful feedback.
> ***Q1: Unclear how efficiently the framework can learn and reuse schemata in continuously evolving contexts.***
>
> We appreciate the point of evaluating the point of real dynamic environments. While there exists no realistic streaming datasets, we have demonstrated the CORRECT’s:
>
> **Robustness and Adaptability**: Figure 6 shows that there is Strong Schema Transferability across Datasets and Models. Therefore, during the cold start phase, we can quickly utilize the previous error schemata from previous datasets and models to guide the initial phases;
>
> **Scalability**: After we gather some initial error schemata of the current dataset, Figure 8 shows that we can have satisfying performance with very small cache sizes: we can improve from the detection accuracies from 14.7->35.4 using only 58 error schemata. Therefore, we can easily adapt our methods in the online scenario by first involving the adaptation from previous caches and then having great performance even with limited cache sizes;
>
> We set up a new experiment where requests arrive continuously online. Specifically, we first initiated the cold start cache with all the error schemata generated using the Who and When dataset. Then, we gradually include more error schemata for the incoming requests for the Hotpot and WikiMQA dataset. The results are also evaluated in the online setting, where we will calculate the performance based on all the current data. As shown below, our method is able to achieve robust performance over time.
>
> | Coverage    | 0–10% | 0–20% | 0–50% | 0–100% |
> |-------------|-------|-------|-------|--------|
> | Hotpot      | 35.8  | 40.6  | 41.2  | 39.4   |
> | WikiMQA     | 70.8  | 68.6  | 67.6  | 69.1   |
>
>
>
>
> ***Q2 & Q3: Additional overhead and dependency may limit scalability in fully automated or large-scale deployments. Future work could explore strategies for automated or semi-supervised schema validation. The idea of using human-verified cache to improve model exists in the literature, eg, EcoAssistant.***
>
> Thanks for this insightful feedback. We are happy to incorporate your feedback and add this discussion to our final version. We agree that human-verified confirmation during cache construction introduces some overhead, and exploring automated or semi-supervised schema validation is a valuable direction for future work. However, we remark that
>
> Human-in-the-loop supervision is not strictly required for CORRECT. In practical deployments (e.g., real conversational systems such as ChatGPT or Gemini), user feedback is already naturally collected at scale through various measures;
>
> A schema can be reused across many queries, and even a small number of error schemata can already greatly enhance the error recognition performance. Thus, the amortized cost per task is small. As demonstrated in our cache-size experiments in Figure 8, only 58 error schemata of the entire cache can improve the performance on HotpotQA from 14.7->35.4. This further reduced the average costs in the real deployments. When human supervision is unavailable, one can also rely on LLM-as-a-judge with confidence filtering to automatically verify successful trajectories before cache insertion.
> We also appreciate the reviewer’s suggestion about leveraging prior work on human-verified caches such as EcoAssistant, and we will include a discussion comparing these approaches and outlining future directions for reducing manual confirmation while preserving interpretability and reliability.

---

### Official Review · Reviewer_m16n · 2025-11-02

**Soundness:** 3
**Presentation:** 3
**Contribution:** 3
**Rating:** 6
**Confidence:** 4

**Summary:**

The paper addresses the challenge of error recognition in MAS execution, finding the root cause of failure in a long trajectory. The paper builds upon the insight that while the specifics of the failure trajectory appear unique, they stem from recurring structurally similar patterns.

With this, the authors' propose CORRECT, a training-free framework, that builds an "error schemata" by offline analysis of failure trajectories, identifying the signature pattern and context of errors. During online inference, it retrieves the most relevant schemata from a cache, providing them to LLM-as-judge, enabling targeted and accurate error localization without fine-tuning.

The paper also introduced CORRECT-Error, a large-scale benchmark of annotated failure trajectories.

**Strengths:**

- The paper proposes a technique that is initialized with offline analysis of failure trajectories, but can also update the schemata/cache online, to identify new failure modes as they occur during deployment
- The paper shows evidence of human alignment with bootstrapped trajectories
- Evaluation across diverse benchmarks and models, demonstrating accuracy improvements in MAS with the intervention

**Weaknesses:**

- The primary baselines are naive ICL, zero-shot LLM-as-judge and finetuning. Since the proposed approach is about context intervention through a cache, could the authors comment on prompt optimization techniques like MIPRO as a baseline?
- The dataset is built by corrupting successful trajectories through error injection. Could the authors' discuss the effect of sampling bias by only selecting from task instances for which an existing MAS is able to successfully bootstrap a valid solution. What about domains with close to 100% failure rates?

**Questions:**

- Could the author's provide examples of schema generated by CORRECT in offline and online phases? While one of the emphasis of CORRECT is inference-time schema inference, could the authors' discuss how is the offline mode different from the application of a static human curated taxonomy like MAST?
- Could the authors' address the impact of starting from successful trajectories, what about domains with high failure rates?

---

> ### Author Response · Authors · 2025-11-21
>
> We thank you for the insightful feedback.
>
> ***W1: Could the authors comment on prompt optimization techniques like MIPRO as a baseline?***
>
> MIPRO (Multi-prompt Instruction Proposal Optimizer) is a Bayesian optimization algorithm that learns optimal instructions and few-shot demonstrations for LLM programs through bootstrapping successful execution traces and discrete search over candidate combinations.
>
> MIPRO is naturally complementary to our thought template approach rather than a competing baseline. Our similarity-based retrieval provides relevant error pattern candidates, while MIPRO optimizes presentation aspects: instruction phrasing and demonstration selection. However, there is a fundamental difference in how examples are selected: MIPRO uses a fixed set of few-shot demonstrations optimized on the training set and applied uniformly to all test cases, whereas our method dynamically retrieves the most relevant error templates for each specific test case based on trajectory similarity. This dynamic, per-instance retrieval is critical for error detection, as different failure cases require different error pattern knowledge. We conducted experiments combining MIPRO with our method on the Hand-Crafted dataset using Qwen-7b. While MIPRO can optimize instruction phrasing, the fixed few-shot examples hurt performance compared to our similarity-based template selection. Our approach of selecting error templates dynamically per test case based on trajectory similarity proved more effective than MIPRO's strategy of using fixed demonstrations across all cases.
>
>
> |                | Correct | MIPRO |
> |----------------|---------|-------|
> | Hand-Crafted   | 12.2    | 8.6   |
>
>
>
>
>
> ***W2 and Q2: The effect of sampling bias by only selecting from task instances for which an existing MAS is able to successfully bootstrap a valid solution. What about domains with close to 100% failure rates?***
> We thank the reviewer for raising the question about sampling bias and domains with extremely high failure rates. We want to remark that the task failure rate is not equivalent to the failure rate of error recognition. CORRECT requires some successful trajectories to construct ground-truth step-level error labels during offline distillation (§3.1), but it does not require the overall task domain to have a high success rate.
> 1. Even in high-failure-rate domains, sufficient successful trajectories can still be obtained.
> In practical MAS deployments that process millions of requests, even a 1% success rate yields tens of thousands of successful trajectories. This is enough for CORRECT to build an initial schemata cache. Moreover, successful examples need not come solely from the same agent: one can (i) sample more runs offline, (ii) query stronger or more compute-heavy models to bootstrap a small number of correct trajectories, or (iii) incorporate minimal human supervision where needed. Our distillation procedure further amplifies this small seed set to produce high-coverage schemata.
> 2. CORRECT is robust with very small numbers of schemata.
> As shown in our cache-size experiment (Fig. 8), the performance saturates quickly. Only 58 error schemata can improve the performance on HotpotQA from 14.7->35.4. This means CORRECT does not require a large volume of successful trajectories to be effective.
> 3. Strong transferability mitigates domain-specific sparsity.
> We demonstrate in Fig. 6 that schemata learned from one dataset/model transfer effectively to others. For instance, the error schema generated on Hand-Crafted dataset can still improve the performance on Algorithm-generated dataset from 31.8 to 36.5. Thus, for domains where collecting successes is difficult, CORRECT can leverage schemata distilled from other models or datasets to perform accurate error recognition.
> 4. Empirical evidence: CORRECT remains effective even when the underlying task has ~100% failure rate.
>
> The WHO&WHEN datasets consist entirely of failed trajectories with human-annotated error steps, which effectively simulate near-100% failure scenarios. As reported in Table 1, CORRECT still delivers substantial performance gains on these subsets. For instance, our method improves the performance of Qwen-7b from 3.5 to 12.1 on Handcrafted dataset and improves the performance of GPT-5 from 15.9 to 38.1 on Algorithm-Generated dataset.
> Thus, while CORRECT requires some successful trajectories to form initial schemata, (i) real-world scale, (ii) cache efficiency, (iii) cross-domain transferability, and (iv) empirical experiments together ensure that CORRECT remains practical and effective even in domains with extremely low MAS success rates.

---

> ### Author Response · Authors · 2025-11-21
>
> ***Q1: Could the author's provide examples of schema generated by CORRECT in offline and online phases? While one of the emphasis of CORRECT is inference-time schema inference, could the authors' discuss how is the offline mode different from the application of a static human curated taxonomy like MAST?***
>
> Thank you for the insightful question. We want to remark that CORRECT generates error schemata asynchronously from the serving pipeline. During online inference, CORRECT simply retrieves the most relevant schemata from the cache with less than 5% overhead to the serving latencies, as shown below. We also have shown the example of our error schema in Figure 4. We will show more sample error schemata in the appendix in the updated version.
>
>
>
> |                | Baseline | Correct (Ours) |
> |----------------|----------|----------------|
> | API cost       | 0.86     | 0.89           |
>
>
>
>
> |                 | Baseline | Correct (Ours) |
> |-----------------|----------|----------------|
> | Inference time  | 44s      | 47s            |
>
>
>
> We further compare and contrast the error schemata generated by CORRECT with  human-curated taxonomies in MAST below:
>
> Error Schemata from CORRECT vs. static human-curated taxonomies (e.g., MAST).
> Compared with CORRECT, Human-curated taxonomies face several limitations:
> Lack of granularity. Human taxonomies operate at coarse conceptual levels (e.g., “wrong tool use,” “missing precondition”), but they do not provide the fine-grained, step-specific patterns required for localizing an error within a trajectory.
>
>
> High manual cost and slow iteration. Curating and updating a taxonomy like MAST requires domain expertise and substantial labor. As models, tools, or task distributions evolve, these taxonomies quickly become outdated.
>
>
> Limited adaptability to distribution drift. Static taxonomies cannot automatically account for new failure modes emerging as environments change, agents are upgraded, or new tasks appear.
>
>
> Overall, CORRECT’s offline schema generation automatically distills fine-grained, step-level schemata from observed trajectories and updates them asynchronously to track the live serving distribution—without adding any overhead to online inference. This continual adaptation enables CORRECT to capture newly emerging failure modes and remain effective under distribution drift, offering a level of precision and practicality that static human-curated taxonomies cannot match in large-scale, real-world deployments.

---

### Author Response · Authors · 2025-11-25

Dear reviewers,
Thank you again for the thoughtful time and effort you’ve dedicated to evaluating our submission. We have now uploaded a detailed rebuttal addressing all questions and concerns raised in the reviews. If you have a moment, we would be grateful if you could take another look and let us know whether our clarifications resolve any of the issues or prompt any updates to your assessment. Please feel free to indicate if any part of our response would benefit from additional explanation.
We sincerely appreciate your consideration.

---

### Author Response · Authors · 2025-11-28

We have incorporated all clarifications and updates from our rebuttal into the revised manuscript. We kindly invite the reviewers to take a look at the updated version and let us know if there are any remaining questions or points that would benefit from further discussion. We would greatly appreciate any updates to your assessments based on the revisions. Thank you again for your time and thoughtful feedback.

---

### Author Response · Authors · 2025-12-02
**Summary of Reviewer Feedback and Corresponding Updates**

**Dear AC and all reviewers,**

We sincerely thank all the reviewers for their thoughtful reviews and invaluable feedback. We truly appreciate the time and effort you devoted to carefully reading our paper and providing constructive comments. Your insights have been extremely helpful in strengthening the clarity, rigor, and presentation of our work.

We are grateful that the reviewers recognized several positive aspects of our framework CORRECT, including its **novelty** in using schema-based knowledge transfer (Reviewer m16n, yo98), its **strong empirical performance** across diverse models(Reviewer m16n, BG9L), and the contribution of the **CORRECT-Error benchmark** to the community(Reviewer bUB3, yo98). We would also like to respectfully highlight to the AC that, before the system interruption, the reviewer who gave a score of 4 had raised it to 6 based on our responses (“Most of my concerns have been addressed” “I have revised my score accordingly”).

For the comments and concerns raised in the reviews, we have prepared individual, ***point-by-point responses for each reviewer to address all the questions***. We have also revised the manuscript accordingly, and we summarize the key changes below. To facilitate cross-referencing, all modifications are highlighted in blue in the updated version.

**Paper Update Summary:**

**1. Additional evaluation and analysis:**

- **Overhead of Schema-Guided Recognition** (Section 5.3 and Table 4), demonstrating that CORRECT introduces minimal overhead (<5% latency increase) and negligible API cost increase (<5% cost increase) . (suggested by Reviewer  BG9L)

- **Sensitivity to Similarity and Hotness Thresholds** (Section 5.3, Table 7 and Table 8), showing robust performance across settings (e.g., ARC accuracy increases from 80.4 to 81.3 as θ_hot increases). (suggested by Reviewer BG9L)

- **Robustness to Retrieval Noise** (Section 5.3, Table 9, and Table 10), utilizing hard-negative experiments to show CORRECT retains performance gains even when irrelevant schemata are injected (e.g., GPT-5 maintains 10.7% vs baseline 8.6% with 5 random schemata). (suggested by Reviewer yo98)

- **Online Deployments / Streaming Evaluation** (Section 5.2 and Table 5), showing CORRECT maintains stable accuracy in continuously evolving environments even when only 10–20% of newly observed schemata are incorporated  (suggested by Reviewer bUB3, Reviewer yo98)

- **Comparison with MIPRO** (Section 5.1 and Table 1), demonstrating that dynamic schema retrieval outperforms fixed optimization strategies. (suggested by Reviewer m16n, Reviewer yo98)

- **Robustness to context-length truncation** (Appendix A.12 and Table 5), showing CORRECT is robust to different context-length choices during retrieval (suggested by Reviewer BG9L)

- **Transferability to error-category settings** (Appendix A.14 and Table 6), showing CORRECT can help to discover the different error-categories as well. (suggested by Reviewer BG9L)

**2. Writing:**

- Update **Section 3** to better clarify the workflow in Algorithm 1, detailing how the offline stage builds the initial cache, the online stage guides the LLM detector, and the dynamic stage refines schemata to keep the cache compact and accurate. (suggested by Reviewer BG9L)
- Update **Section 5.2** to discuss how the framework adapts to dynamic MAS deployments and evolving request distributions. (suggested by Reviewer bUB3, yo98)
- Add **Appendix A.7** to include detailed overhead profiling (latency and API cost). (suggested by Reviewer BG9L)
- Add **Appendix A.8** to discuss the comparison between CORRECT schemata and static human-curated taxonomies. (suggested by Reviewer m16n)
- Add **Appendix A.9** to discuss human verification, automated validation, and scalability. (suggested by Reviewer bUB3)
- Add **Appendix A.10** to include detailed implementation settings and hyperparameters for the Fine-Tuning (SFT) baseline. (suggested by Reviewer BG9L)
- Add **Appendix A.11** to include the full prompts used for natural error injection. (suggested by Reviewer BG9L)
- Add **Appendix A.12** to demonstrate robustness to context-length truncation. (suggested by Reviewer BG9L)
- Add **Appendix A.13** to analyze failure cases, specifically multi-error trajectories. (suggested by Reviewer BG9L)
- Add **Appendix A.14** to demonstrate transferability to error-category settings. (suggested by Reviewer BG9L)
- Add **Appendix A.15** to provide sensitivity analysis for similarity δ and hotness θ_hot  thresholds. (suggested by Reviewer BG9L)
- Add **Appendix A.16** to detail the robustness to retrieval noise experiments. (suggested by Reviewer yo98)

We thank all the reviewers for their constructive feedback, which has helped us improve the paper. We hope that our responses and the revised paper address the reviewers’ concerns, and we are happy to answer any further questions.

Best regards,

**Authors of CORRECT**

---

### Meta-Review · Area_Chair_9onX · 2025-12-23

**Summary:**

This paper introduces CORRECT, a training-free framework for error recognition in Multi-Agent Systems that distills failure trajectories into reusable error schemata and retrieves them at inference time to guide error localization. The reviewers acknowledged several strengths including the novelty of schema-based knowledge transfer, strong empirical gains across diverse models and benchmarks, and the contribution of the CORRECT-Error benchmark. However, significant weaknesses remain. The framework critically depends on highly capable proprietary models for offline schema extraction, raising concerns about accessibility and reproducibility. The baseline comparisons are limited, focusing primarily on naive approaches rather than specialized verification pipelines. The benchmark construction, which uses schema-guided error injection, may introduce bias that artificially inflates the method's advantages on synthetic data, and while WHO&WHEN provides some natural evaluation, the extent to which real failures follow assumed reusable patterns remains unclear. Presentation issues were noted, with technical details initially sparse or relegated to appendices, though authors promised revisions. The decision to reject stems primarily from concerns about the method's practical deployability given its reliance on expensive proprietary models, insufficient comparison with relevant baselines, and questions about whether the evaluation adequately captures the diversity of real-world failure modes.

**Reviewer Concerns:**

The authors provided extensive responses including new experiments on overhead, sensitivity analysis, robustness to retrieval noise, online deployment scenarios, and MIPRO comparisons. Reviewer BG9L explicitly stated that most concerns were addressed and raised the score from 4 to 6, noting satisfaction after seeing promised revisions incorporated. Reviewer m16n's concerns about MIPRO and sampling bias were partially addressed through new experiments showing CORRECT outperforms MIPRO and demonstrating transferability across datasets. Reviewer bUB3's streaming deployment concern was addressed with new online experiments showing stable performance with limited schema coverage. Key unresolved issues include the fundamental dependence on GPT-5 for schema quality, the computational opacity of schema distillation at scale, and whether gains primarily reflect benchmark artifacts rather than genuine advances in handling diverse real-world failures. The lack of comparison with causal replay methods and specialized verification pipelines remains a gap, though authors argued these are impractical. The decision weighs the acknowledged presentation improvements and additional experiments against persistent concerns about generalizability, baseline coverage, and the practical viability of deploying a system requiring expensive proprietary models for its core functionality.

**Reviewer Scores:**

Regarding score predictions, Reviewer m16n would likely maintain their score of 6 given that their main concerns were addressed with new experiments and clarifications. Reviewer bUB3 would likely stay at 6 as the streaming experiments and discussion of human supervision overhead addressed their primary concerns. Reviewer BG9L would likely update from 4 to 6 after rebuttal. Reviewer yo98 would likely remain at 4, as while extensive experiments were added, fundamental concerns about dependence on proprietary models and potential benchmark bias were acknowledged but not fully resolved, and the reviewer expressed these as core architectural limitations rather than presentation issues.

---

### Decision · Program_Chairs · 2026-01-26

Reject